*Report*

# The phosphatase inhibitor LB-100 creates neoantigens in colon cancer cells through perturbation of mRNA splicing

Matheus H Dias[1,6], Vladyslava Liudkovska [2,6], Jasmine Montenegro Navarro [3], Lisanne Giebel[3], Julien Champagne [3], Chrysa Papagianni [1], Onno B Bleijerveld [4], Arno Velds [5], Reuven Agami [3], René Bernards [1✉] & Maciej Cieśla [2✉]

## Abstract

**Perturbation of protein phosphorylation represents an attractive approach to cancer treatment. Besides kinase inhibitors, protein phosphatase inhibitors have been shown to have anti-cancer activity. A prime example is the small molecule LB-100, an inhibitor of protein phosphatases 2A/5 (PP2A/PP5), enzymes that affect cellular physiology. LB-100 has proven effective in pre-clinical models in combination with immunotherapy, but the molecular underpinnings of this synergy remain understood poorly. We report here a sensitivity of the mRNA splicing machinery to phosphorylation changes in response to LB-100 in colorectal adenocarcinoma. We observe enrichment for differentially phosphorylated sites within cancer-critical splicing nodes of U2 snRNP, SRSF and hnRNP proteins. Altered phosphorylation endows LB-100-treated colorectal adenocarcinoma cells with differential splicing patterns. In PP2A-inhibited cells, over 1000 events of exon skipping and intron retention affect regulators of genomic integrity. Finally, we show that LB-100-evoked alternative splicing leads to neoantigens that are presented by MHC class 1 at the cell surface. Our findings provide a potential explanation for the pre-clinical and clinical observations that LB-100 sensitizes cancer cells to immune checkpoint blockade.**

**Keywords** Alternative Splicing; LB-100; Colorectal Adenocarcinoma; Combination Therapies in Cancer; Neoantigen Formation
**Subject Categories** Cancer; Immunology; RNA Biology

## Introduction

Protein phosphatases act in a multitude of cellular signaling pathways and are commonly deregulated in cancer (Stanford and Bottini, 2023). Previous efforts have identified LB-100 as a dual inhibitor of protein phosphatases 2A (PP2A) and 5 (PP5) (D'Arcy et al, 2019). Both phosphatases have a reported function in cancer (Kauko et al, 2020; Vervoort et al, 2021; Zhang et al, 2023), with PP2A considered a tumor suppressor gene given its potential to modulate various cancer-related signaling pathways and thereby inhibiting tumor development (Arroyo and Hahn, 2005; Janssens and Goris, 2001; Junttila et al, 2007). Drug development efforts have therefore focused on small molecule activators of PP2A, as such drugs could potentially act on multiple cancer-relevant signaling routes to suppress tumorigenicity. Surprisingly, inhibition of PP2A/PP5 with LB-100 has also exhibited anti-cancer effects, especially when used in combination with radiotherapy or specific chemotherapy drugs (O'Connor et al, 2018). This effect is believed to result from the notion that PP2A inhibition affects DNA repair mechanisms, making cells more vulnerable to DNA-damaging agents (Lv et al, 2014; Wei et al, 2013). Our recent data show increased oncogenic signaling in response to LB-100, causing DNA replication stress, which may also contribute to a greater sensitivity to DNA-damaging agents (Dias et al, 2024). An independent and potentially powerful anti-cancer effect of PP2A/PP5 inhibition stems from the finding that LB-100 also enhances the efficacy of immune checkpoint blockade in different cancer models by fostering T-cell and cGAS-STING activation (Apostolidis et al, 2016; Ho et al, 2018; Maggio et al, 2020). Moreover, LB-100 has been shown to turn immunologically "cold" tumors (i.e. those poorly recognized by the immune system) "hot" by inhibiting DNA mismatch repair (Yen et al, 2021). Such "promiscuous" sensitization of PP2A/PP5 inhibition to different therapeutic modalities reflects their central role in controlling cancer cell homeostasis.

One of the emerging therapeutic strategies for cancer treatment is to leverage tumor-associated epitopes presented in the context of Major Histocompatibility Complex Class I (MHC-I) (Lang et al, 2022; Schumacher and Schreiber, 2015). These non-canonical immunopeptides can activate the immune system against cancer cells through recognition by autologous T cells (Lang et al, 2022).

[1]Division of Molecular Carcinogenesis and Oncode Institute, The Netherlands Cancer Institute, Amsterdam, The Netherlands. [2]IMol Polish Academy of Sciences, Warsaw, Poland. [3]Division of Oncogenomics and Oncode institute, The Netherlands Cancer Institute, Amsterdam, The Netherlands. [4]Proteomics Facility, Netherlands Cancer Institute, Amsterdam, Netherlands. [5]Central Genomics Facility, Netherlands Cancer Institute, Amsterdam, Netherlands. [6]These authors contributed equally: Matheus H Dias, Vladyslava Liudkovska. ✉E-mail: r.bernards@nki.nl; m.ciesla@imol.institute

The newly formed neoepitopes, which are not immune-privileged and are expressed exclusively outside the healthy tissues, create an attractive window of opportunity for anti-cancer vaccines or checkpoint immunotherapies (Lang et al, 2022). The cancer-specific formation of neoantigens has been reported to stem from somatic mutations and genetic alterations (Lang et al, 2022), translation from a novel or non-canonical open reading frames (Ouspenskaia et al, 2022), and altered function of splicing components (Bigot et al, 2021; Lu et al, 2021). Regarding the latter, various cancers have been found to harbor mutations in the core splicing machinery (Wang and Aifantis, 2020). It is now acknowledged that alternative splicing can be a source of tumor neoantigens, and there are ongoing efforts to develop pharmacological treatments that target this process (Lee and Abdel-Wahab, 2016; Seiler et al, 2018). Accordingly, a recent study indicates that pharmacological modulation of splicing drives the formation of functional neoantigens to elicit anti-tumor immunity, inhibiting tumor growth, and enhancing checkpoint blockade in a manner dependent on host T cells (Lu et al, 2021). Collectively, these data suggest that splicing modulators are attractive targets for combinatorial therapies alongside immune checkpoint blockade approaches.

Given the growing body of evidence supporting the anti-cancer effects of LB-100, particularly its capability to enhance tumor immune sensitization (Ho et al, 2018), understanding the global impact of this phosphatase inhibitor on the phosphoproteome of cancer cells could provide further mechanistic insights into why cancer cells are susceptible to LB-100. Moreover, such studies could suggest potent drug combinations by identifying specific processes that are perturbed by the drug. Here, we addressed the impact of LB-100 on the phosphoproteome of colorectal cancer cells to infer the most relevant cellular processes altered by the drug. Our results identify a major role for LB-100 in perturbing the proper execution of RNA splicing. We show that alternative splicing yields neoantigen formation on cancer cells, supporting the rationale for combining LB-100 with checkpoint immunotherapies.

## Results and discussion

### LB-100 drives changes in the phosphoproteome of splicing regulators

To delineate how the protein repertoire of cancer cells is affected by LB-100, we treated human colorectal adenocarcinoma cells SW-480 with LB-100 for 12 h and performed paralleled quantitative mass spectrometry-based proteomics and phosphoproteomics (Fig. 1A). Comparative analysis of LB-100-treated samples with untreated controls revealed relatively few changes in protein abundance, with 120 and 239 proteins being respectively down- and up-regulated in response to PP2A inhibition (Fig. 1B). In contrast, phosphoproteomic analysis revealed widespread changes in protein phosphorylation status, with >1700 proteins undergoing differential phosphorylation in LB-100-treated cells (Fig. 1C). Consistently with the inhibition of phosphatase activity by the drug, we observed a bias toward hyperphosphorylation events (62%, encompassing 2099 significantly affected phosphosites from 1069 proteins), as compared to hypophosphorylation events (38%, comprising 1364 affected phosphosites from 649 proteins) in LB-100-treated cells.

These phosphorylation events encompassed primarily serine/threonine residues (3452 out of 3463, 99.68% affected phosphosites). Moreover, differentially phosphorylated proteins from our dataset strongly overlapped with previously reported targets of PP2A from studies using genetic PP2A inhibition (Kauko et al, 2020) or cell-based dephosphorylation assay (Hoermann et al, 2020) from HeLa cells, highlighting this phosphatase as a prime target of LB-100 in colorectal carcinoma (Fig. EV1A) (Hoermann et al, 2020; Kauko et al, 2020).

We reasoned that the widespread changes in phosphorylation downstream to LB-100 may affect disparate biological processes important for cancer cell viability. Indeed, analysis of proteins exhibiting increased (Fig. 1D) and decreased (Fig. 1E) phosphorylation upon treatment revealed enrichment for pathways critical to cancer, including cell cycle progression, signal transduction, and chromatin remodeling (Fig. 1D,E). Also, there was a notable group of hyperphosphorylated proteins enriched for mitotic regulators, including PCNT and AURKB, reported to play a key role in cancer progression. Interestingly, among the most overrepresented biological processes in both hyper- and hypophosphorylated groups, we detected numerous terms linked to RNA metabolism, with enrichment for proteins related to mRNA splicing regulation and control (Fig. 1D–F). In fact, approximately 39% (125 out of 315) of human splicing regulators were differentially phosphorylated in response to LB-100 treatment. We observed significant enrichment for all categories of phosphorylation events (hyper-, hypo-, and biphasic phosphorylation encompassing splicing regulators in which different amino acid residues underwent increased or decreased phosphorylation), rather than skewing for hyperphosphorylation events observed for the other groups of aberrantly phosphorylated proteins (Fig. 1F).

The spliceosome is a megadalton, multimodal RNA-protein machinery consisting of five small nuclear ribonucleoprotein complexes and a multitude of transiently interacting splicing regulators that can both activate and repress alternative splicing (Will and Lührmann, 2011). Projection of differentially phosphorylated splicing factors on a map of protein–protein interactions within the spliceosome unveiled specific groups of splicing regulators sensitive to phosphorylation changes induced by LB-100 (Fig. 1G). These phosphorylation changes partially overlap with reported (Hoermann et al, 2020; Kauko et al, 2020) targets of PP2A (Fig. EV1B,C), including different groups of splicing factors (Fig. EV1D–F), and were largely uncoupled from differences in total protein abundance (Fig. 1G). This is in line with previous reports that protein phosphorylation affects function and protein–protein interactions rather than expression of the differentially phosphorylated proteins (Nishi et al, 2014). We observed that multiple constituents of the core complex of U2 small nuclear ribonucleoprotein (snRNP) underwent differential phosphorylation events in response to LB-100. U2 snRNP is recruited to the intron branch site to form the A complex, and then, upon recruitment of U4/U6.U5 tri-snRNP, it forms the B and B$^{act}$ splicing complexes (Will and Lührmann, 2011). In this way, U2 snRNP binding confers a rate-limiting and fidelity-ensuring stage of the reaction that enables nucleophilic attack of a branch point adenosine on the 5' splice site. We observed that U2 components previously described as key cancer fate determinants (Cieśla et al, 2023; Lee and Abdel-Wahab, 2016), such as RBM8A, splicing factor 3B subunit 1 (SF3B1), or U2AF1 were among mis-phosphorylated

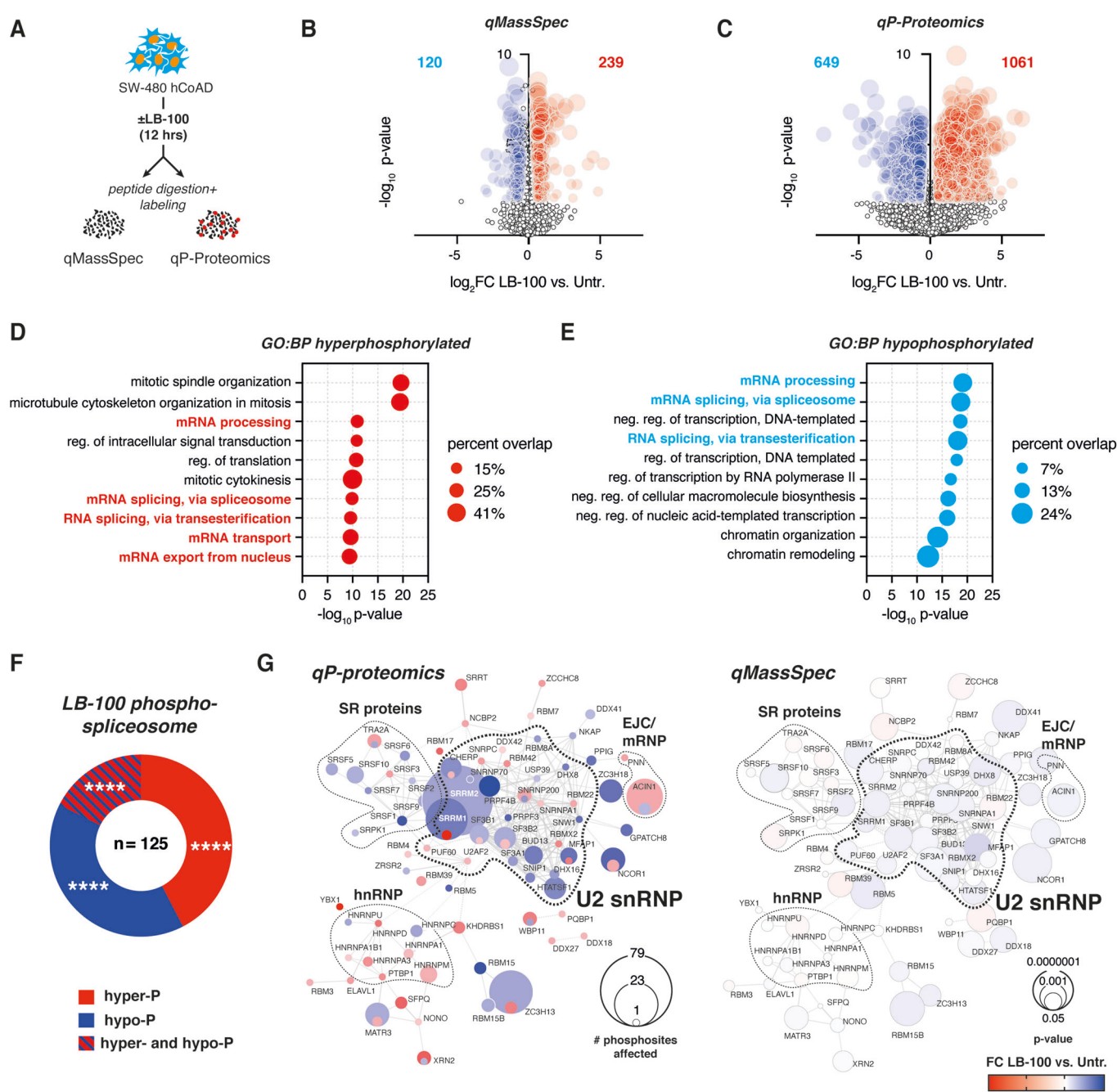

**Figure 1. PP2A/PP5 small drug inhibition by LB-100 drives phosphoproteome changes enriched for multiple components of the spliceosome.**

(A) Schematic illustrates an experimental approach to determine proteomic (quantitative mass spectrometry, qMassSpec) and phosphoproteomic (qP-Proteomics) changes in response to PP2A/PP5 inhibitor LB-100 in human colorectal adenocarcinoma (hCoAD) cell line SW-480. (B) Volcano plot shows protein expression changes in response to 4 μM LB-100 stimulation (12 h) compared to untreated controls (Untr.). Upregulated (red) and downregulated (blue) proteins with false discovery rate (FDR) <0.05 (t-test) and |Log₂FoldChange| >0.58 are shown. (C) Volcano plot shows phosphorylation changes in response to 4 μM LB-100 stimulation (12 h) compared to untreated controls. Hyper- (red) and hypophosphorylated (blue) events with FDR <0.05 (t-test) and |Log₂FC| >0.58 are shown. (D) Gene Ontology (GO) analysis for 1061 hyperphosphorylated proteins. BP biological process. (E) GO analysis for 649 hypophosphorylated proteins. (F) The Donut chart shows the distribution of proteins with significantly changed (FDR <0.05 and |Log₂FC| >0.58) phosphorylation status. (G) Interaction maps of spliceosome components hyper- and hypophosphorylated (left) in response to 12 h of LB-100 stimulation in SW-480 cells. Corresponding analysis of protein expression changes is shown (right). Connecting lines show interactions from STRINGdb. Data information: In (D, E) p values were calculated with Fisher's exact test. In (F) ****p < 0.0001 (hypergeometric test vs. background). See also Fig. EV1 and Datasets EV1, EV2.

proteins (Fig. 1G). Moreover, we found that many of the differentially phosphorylated proteins belong to the serine-arginine-rich splicing factor family (SRSF) and heterogenous nuclear RNPs (hnRNPs) (Fig. 1G and EV1E). SRSFs and hnRNPs are generally recognized as, respectively, activators and repressors of the splicing and key determinants of cancer progression through concerted control of cancer-initiating alternative splicing events (Urbanski et al, 2022). Proteins such as SRSF1 or SRSF2, which we report to be mis-phosphorylated in LB-100-treated cells, are known cancer regulators (Urbanski et al, 2022). Of note, for all of the four splicing proteins recurrently mutated in cancer (SRSF2, U2AF1, SF3B1, and ZRSR2), we observed changes in the phosphorylation patterns in response to LB-100. Mutations often result in abnormal activity of these splicing factors and altered splicing patterns (Lee and Abdel-Wahab, 2016), contributing to the oncogenic phenotype. However, how the phosphorylation of these factors might change their function during cancer progression and in response to targeted therapies remains largely unexplored.

## LB-100 promotes alternative splicing in colorectal adenocarcinoma cells

Motivated by the observations that phosphorylation critically tunes protein–protein interactions and affects protein activity, we reasoned that the observed rewiring of splicing factors phosphorylation might affect splicing outcomes in colorectal adenocarcinoma cells treated with LB-100. To address this, we performed paired-end RNA-sequencing (RNA-seq) in control and LB-100-treated cells at the time point matching our prior quantitative phosphoproteomic analysis. To rule out possible cell line-specific effects, we performed sequencing in two different human colorectal adenocarcinoma cell lines having different driver oncogenetic backgrounds (SW-480-KRAS$^{G12V}$ and HT-29-BRAF$^{V600E}$, Fig. 2A). Analysis of independent duplicate RNA-seq experiments using the rMATS pipeline (Shen et al, 2014) identified reproducible alterations within ~2000 alternative splicing events (ASEs) enriched predominantly for exon skipping (SE) and intron retention (IR) (Fig. 2B). Using delta percent spliced-in (ΔPSI) as a metric for splicing efficiency, we observed that LB-100 treatment led to a significantly higher number of ASE. That resulted in increased inclusion of alternative exons and aberrant intron retention as compared to control cells (Fig. 2B,C). This entails that LB-100 treatment favors alternative splicing and impacts ASE patterns by abnormal functionality of core splicing machinery. That could be achieved either by boosting the function of splicing activators that recognize specific regulatory motifs in alternative exons with increased splicing efficiencies, or by decreasing the activity of negative regulators. This is in line with our observations on the enrichment of SRSF and hnRNP splicing activators/inhibitors among differentially phosphory-lated proteins in response to LB-100 (Figs. 1G and EV1E).

Specific regulatory motifs within alternatively spliced mRNAs may be determinants of splicing outcomes in response to the changed activity of spliceosome. In line with that notion, differentially spliced exons in LB-100-treated cells shared specific features contributing to alternative exon usage, such as their short length or increased distance from branch point adenosine to the 3'splice site (Fig. 2D). Instead, for introns with changed retention scores in response to LB-100 treatment, we observed differences in length and GC content, features classically associated with the difficult-to-recognize intervening sequences (Fig. 2E). Collectively,

these data indicate that LB-100 results in widespread changes in the phosphorylation status of critical splicing factors and concomitant alternative recognition of introns and exons in human colorectal carcinoma cells.

## LB-100-dependent alternative splicing decreases protein expression of resulting transcripts

Alternative splicing impacts the fate of resulting transcripts in various ways, ranging from changed stability, formation of premature termination codon (PTC), or differential translation rates resulting in changed protein levels, or creation of alternative protein isoform that would differ in its localization, stability, or function (Ule and Blencowe, 2019). To understand the implications of ASEs on gene expression in response to LB-100, we first sought to determine how alternative splicing events downstream of PP2A/PP5 inhibition alter the levels of resulting proteins. We observed a significant decrease in protein levels of transcripts produced from alternatively spliced mRNAs compared to the background proteins (Fig. 2F). Detailed analysis revealed that this was caused primarily by the group of transcripts with differential exon skipping type of ASE and not by the other groups (Figs. 2F and EV2A–D).

Decreased protein levels resulting from ASEs may be mediated by the inclusion of poison exons harboring STOP codons, by the formation of PTC through splicing-in an out-of-the-frame exon, by the formation of alternative 5' or 3' untranslated regions (UTRs), or by the inclusion of protein domains with putative ubiquitination sites. While the latter two are predicted to change the protein expression through post-transcriptional control, without concurrent changes in mRNA levels, the former two could result in accompanying drops in transcript levels. Therefore, to address the level at which LB-100 stimulation affected the protein abundance of AS transcripts, we performed a time-resolved analysis of gene expression by deep sequencing of mRNAs harvested at different time points after PP2A/PP5 inhibition (Figs. 2G and EV2E). Critically, we observed that none of the ASE types resulted in decreased abundances of mRNAs undergoing differential splicing events. Rather, some types lead to paradoxically elevated levels of alternatively spliced mRNAs (Figs. 2G and EV2E,F). Thus, we concluded that the observed effects of LB-100 on alternative splicing and decreased expression of resulting proteins are unlikely to be mediated by transcriptional control. Rather, these observations entail that LB-100 impacts levels of proteins produced on templates of alternatively spliced mRNAs through post-transcriptional regulation, the precise nature of which will be a subject of future studies.

We next hypothesized that the changed stability of alternatively spliced transcripts may be an efficient way to offset protein levels of specific cellular regulators in response to LB-100. Notably, analyses of the skipped exon events revealed a strong enrichment for biological processes related to the maintenance of genomic integrity, including DNA repair, response to DNA damage checkpoint signaling, and homology-dependent repair (Fig. 3A). These observations remained consistent across two different human colorectal adenocarcinoma cell lines (Fig. 3A) and are in agreement both with our previous findings (Cieśla et al, 2023) and those from others studying SF3B1-inhibited human cells (Han et al, 2022). As an example, we observed ASE in transcripts encoding members of the DNA damage response pathway, e.g., ATM, Chk2, and PARPBP (Figs. 3B and EV3), and confirmed that the levels of ATM and Chk2 were reduced in SW-480 cells (Fig. 3C). These

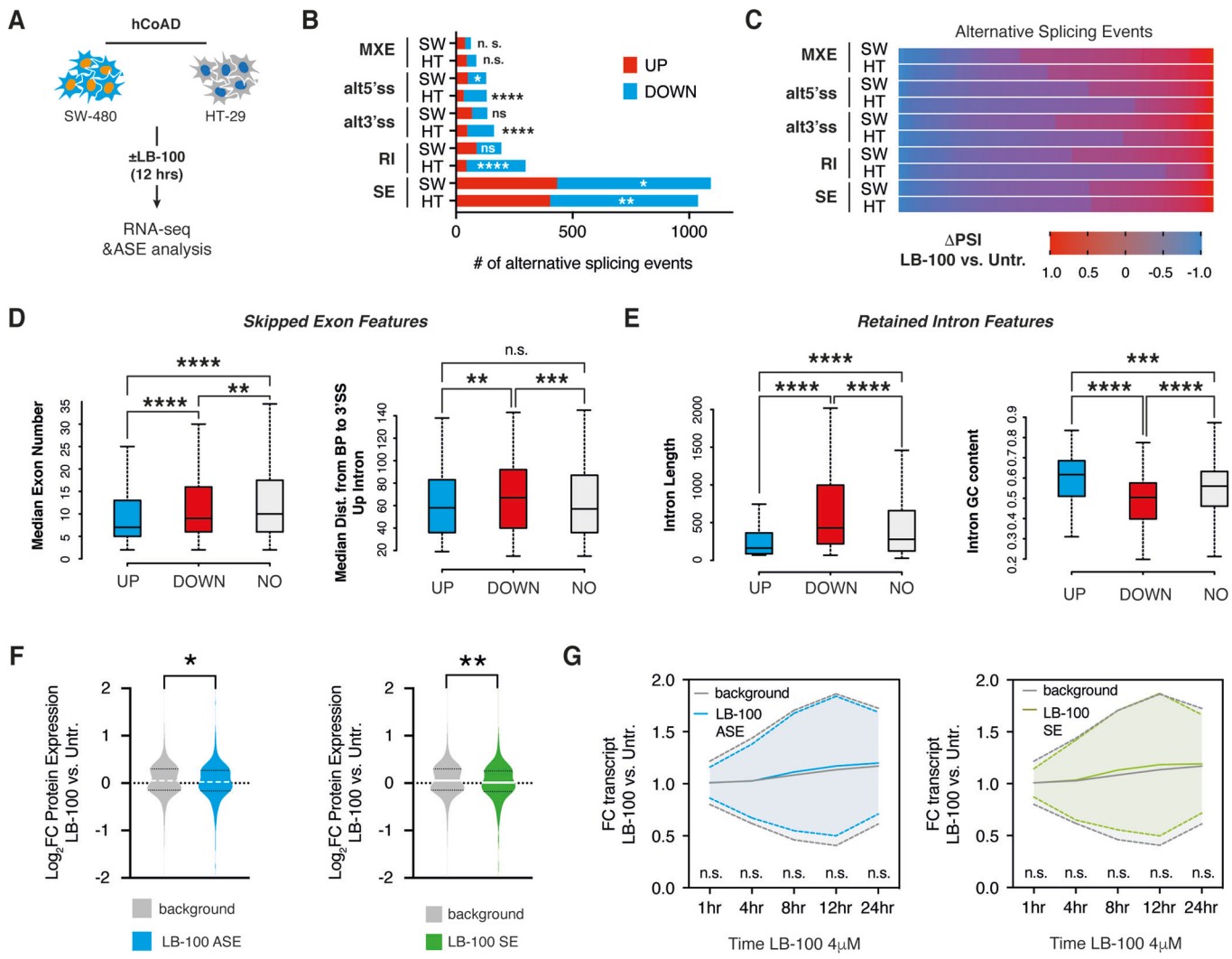

**Figure 2. Selective inhibition of PP2A/PP5 impacts alternative splicing patterns in colorectal adenocarcinoma.**

(A) Experimental setup employed to capture splicing changes in response to PP2A/PP5 inhibitor LB-100 in human colorectal adenocarcinoma (hCoAD) cell lines SW-480 and HT-29. (B) The bar graph shows numbers of skipped exons (SE), mutually exclusive exons (MXE), alternative 3′ splice sites (alt3SS), alternative 5′ splice sites (alt5SS), and retained introns (RI) in LB-100-treated SW-480 and HT-29 cells. (C) Heat map illustrates the quantitative extent of delta percent spliced-in (ΔPSI) for alternative splicing events significantly mis-spliced in response to LB-100 stimulation (FDR <0.05 and |ΔPSI| >0.1). (D) Box-and-whisker plots showing significant differences in median exon number and distance of Branch Point (BP) (Bunda et al, 2015) to 3′splice site (3′ss) in upstream intron in exons with decreased (*n* = 641) or increased (*n* = 409) percent spliced-in values. (E) Box-and-whisker plots showing significant differences in median intron length and guanine-cytosine (GC) content in introns with decreased (*n* = 105) or increased (*n* = 87) percent spliced-in values. (F) The violin plot shows decreased levels of proteins resulting from all alternatively spliced transcripts (left) and in the exon skipping group of ASE (right) in 4 µM LB-100-treated SW-480 cells compared to control in *n* = 4 biological replicates. (G) Graphs show time-resolved changes in all alternatively spliced transcripts (left) and mRNAs from the skipped exon group (right) in response to LB-100 stimulation. Data information: In (B) hypergeometric test was used to calculate ****p < 0.0001; ***p < 0.001; **p < 0.01 in up- vs. down-spliced mRNAs in each family. In (D, E), the box demarcates the first and third quartiles, with the median value shown as a solid line. The boundary of the lower whisker is the minimum value of the dataset, and the boundary of the upper whisker is the maximum value of the dataset. ****p < 0.0001; ***p < 0.001; **p < 0.01 (Mann–Whitney *U*-test). In (F) **p < 0.01; *p < 0.05 (*t*-test). In (G) mean expression is shown as a solid line ± SD depicted as dotted lines. See also Fig. EV2 and Datasets EV3, EV4. Source data are available online for this figure.

results support mechanistically the findings describing that LB-100 sensitizes cancer cells to DNA-damaging agents (Lv et al, 2014).

## Targets of splicing-modulating therapies are differentially phosphorylated in response to LB-100

Phosphorylation of splicing factors was reported previously to be essential for the progression of the splicing reaction (Mermoud

et al, 1994). This includes phosphorylation of SF3B1 at the threonine residues within the N-terminus of the protein. Particularly, a recent study determined CDK11 as a kinase responsible for SF3B1 phosphorylation and transition from B to Bᵃᶜᵗ stage of the splicing (Hluchý et al, 2022). This transition renders the spliceosome catalytically competent for the first transesterification reaction. Additionally, PP1 and PP2A were shown to critically mediate the dephosphorylation of SF3B1,

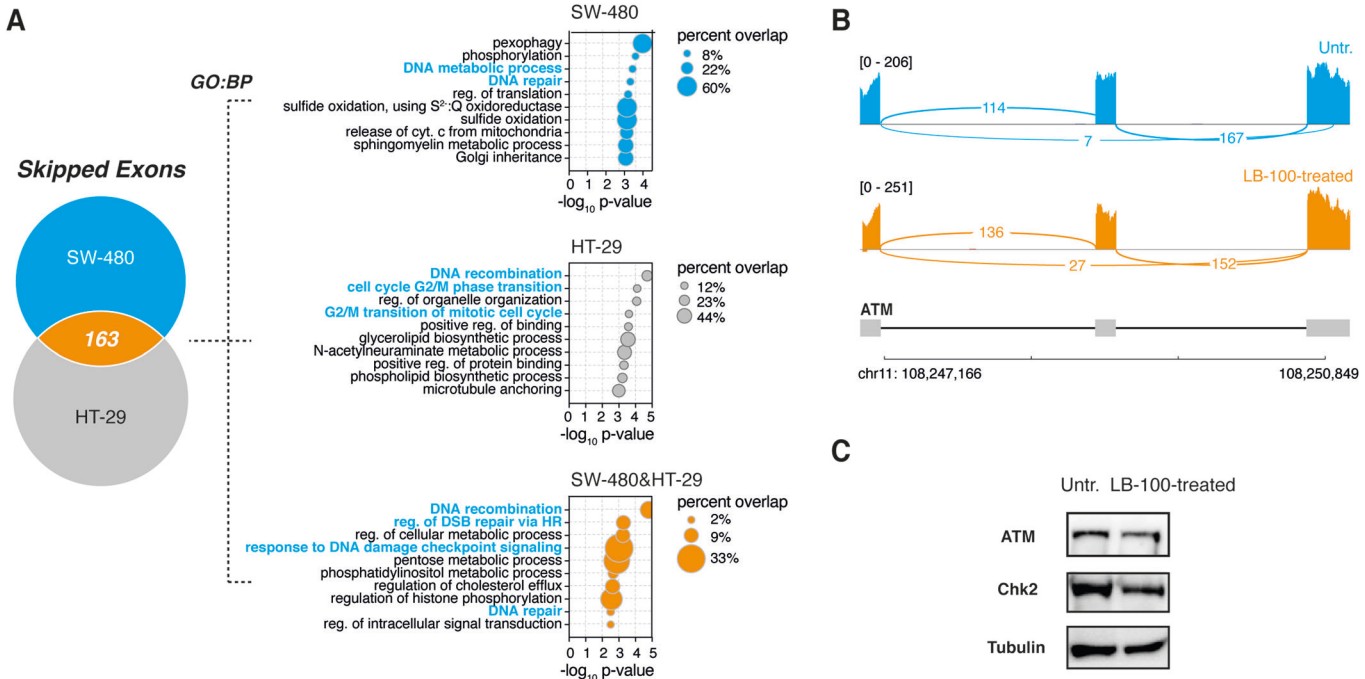

**Figure 3. LB-100-sensitive alternative splicing impacts DNA damage-related mRNAs.**

(A) Venn diagram shows an overlap between SE in LB-100-treated SW-480 and HT-29 cells (left). Gene Ontology (GO) analysis of SE events in SW-480 and HT-29, and an overlap is shown (right). BP biological process. (B) Alternative splicing event within ATM gene and representative Sashimi plots showing differential splicing in untreated and LB-100-treated SW-480 cells. (C) Representative protein analysis of ATM and Chk2 in LB-100-treated SW-480 cell line of human colorectal adenocarcinomas. Data information: In (A) p values were calculated with Fisher's exact test. See also Fig. EV3. Source data are available online for this figure.

essential for progression to the second catalytic step of splicing (Shi et al, 2006). For these reasons, SF3B1 is a primary target of splicing-inhibiting drugs proposed as possible anti-cancer therapies. Drugs targeting SF3B1, such as pladienolides or the orally available H3B-8800, are proposed to drive alternate functionality of the SF3 complex, resulting in changed splicing patterns in particularly vulnerable cancer cells (Lee and Abdel-Wahab, 2016). Our analysis revealed that LB-100 treatment affected phosphorylation within the threonine-proline-rich region, with Ser217, Thr227, and Thr326/328 undergoing hypophosphorylation in response to the drug (Fig. 4A). Notably, Thr328 is one of the residues previously reported to drive the transition to the B^act stage (Hluchý et al, 2022), and all observed differential phosphorylation events were grouped close to the RBM39 protein–protein interaction domain (Fig. 4A). Curiously, RBM39 is the target of indisulam, a molecular glue, and ubiquitin ligase modulator that marks RBM39 for degradation and is a subject of ongoing anti-cancer studies (Han et al, 2017). Our results show that this splicing factor underwent differential phosphorylation at Ser217 within the N-terminus and Ser337 within the activating domain/ESR proteins interaction region upon LB-100 treatment (Fig. 4B). This set of results indicates that various potential anti-cancer targets within the spliceosome are differentially phosphorylated in response to LB-100 treatment.

Phosphorylation often changes the functionality and assembly of protein complexes. To shed light on how the observed phosphorylation changes in spliceosome constituents affected their function, we performed motif enrichment analysis for the

binding of these complexes along differentially spliced mRNAs. We observed qualitative differences in motif enrichment for SF3B4, a protein acting within the same complex as SF3B1, in between up- and down-spliced mRNAs in LB-100-treated cells (Fig. 4C). Moreover, comparative analysis revealed a strong overrepresentation of transcripts sensitive to LB-100 treatment among mRNAs alternatively spliced upon knock-down of SF3B1 under oncogenic stress (Cieśla et al, 2023) as well as among ASEs from previously published datasets for genetic and pharmacological inhibition of RBM39 (Lu et al, 2021) (Fig. 4D). These observations may span beyond these candidates splicing factors, being applicable also to other differentially phosphorylated splicing factors, including members of the SRSF and hnRNP families (Fig. EV4A–G).

We next asked whether LB-100 would be synergistic with known splicing modulators in colorectal adenocarcinoma cells. The results show modest synergy between LB-100 and SF3B1/RBM39 inhibitors (pladienolide B and indisulam, respectively) in SW-480 cells (Fig. 4D). This effect was instead absent in the case of combined treatment with LB-100 and isoginkgetin, an inhibitor of the spliceosome node not affected by LB-100 stimulation (Fig. EV4H). No synergy was observed in HT-29 cells (Fig. EV4I–K), in which we also did not observe positive effects of the combination of pladienolide B and indisulam (Fig. EV4L,M).

Collectively, our results propose that phosphorylation changes in splicing factors expose a selective spliceosome vulnerability in PP2A/PP5-inhibited cancer cells.

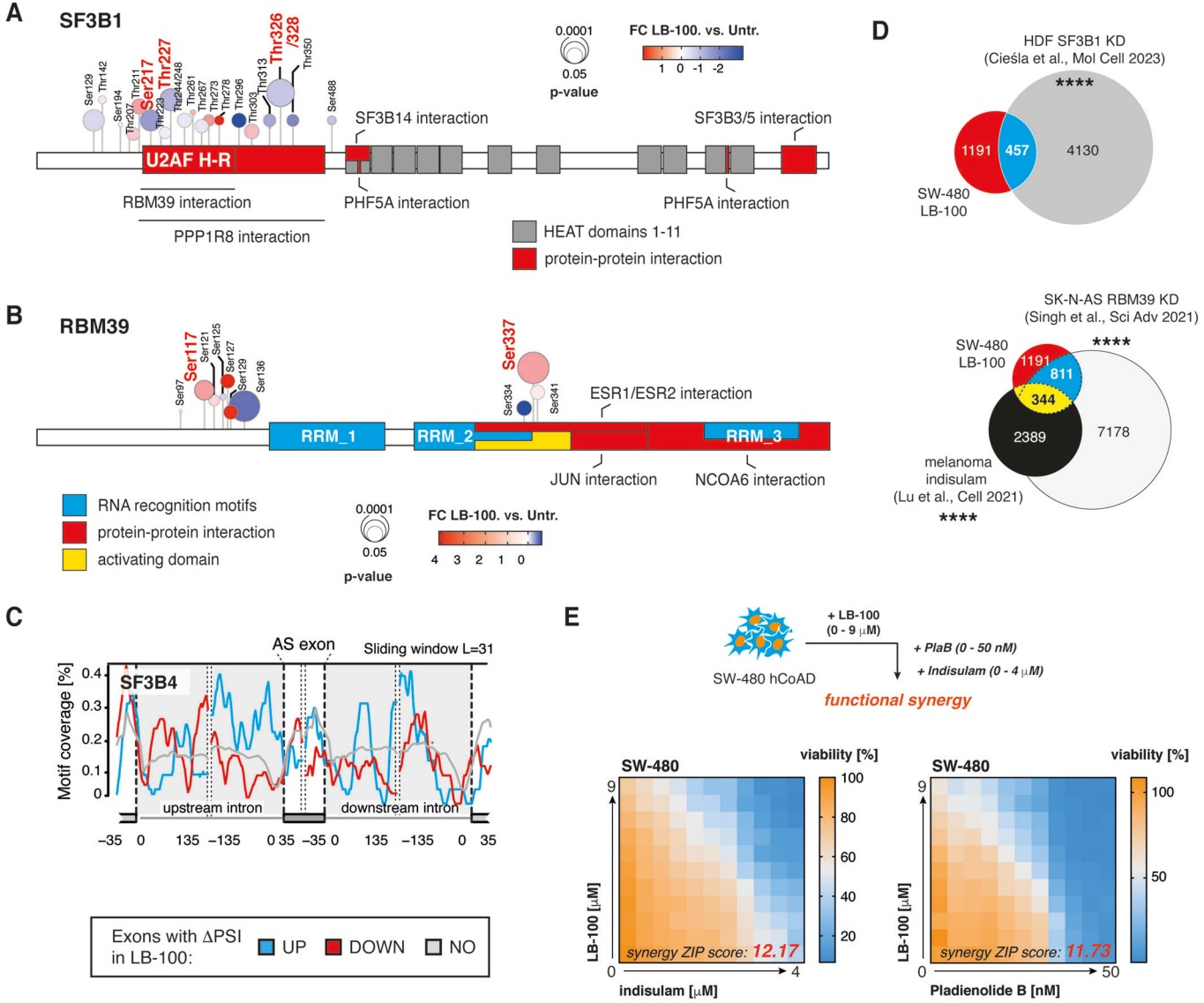

**Figure 4. Targets of splicing-targeting drugs are differentially phosphorylated in response to PP2A/PP5 inhibition.**

(A,B) Location of mis-phosphorylated sites across SF3B1 (A) and RBM39 (B) proteins. Functionally relevant protein regions according to UniProt are shown. Significantly changing phosphosites in response to LB-100 are highlighted in red. (C) Motif RNA map shows SF3B4 binding in the proximity of excluded (DOWN), included (UP), or not affected (NO) exons in LB-100-treated as compared to untreated SW-480 colorectal adenocarcinoma cells. (D) Venn diagram shows a significant overlap between AS transcripts in LB-100-treated SW-480 cells, SF3B1-depleted MYC-expressing Human Diploid Fibroblasts (HDF) (Cieśla et al, 2023), indisulam-treated melanomas (Lu et al, 2021), and RBM39 knock-down (KD) SK-N-AS neuroblastoma cell line (Singh et al, 2021). (E) Synergy matrices show cooperativity between LB-100 and pladienolide B and indisulam in the SW-480 human adenocarcinoma cell line. Representative heat map for $n = 3$ biological replicates. A schematic representation of the experiment is shown on top. Data information: In (D) statistical significance was calculated with hypergeometric test, ****$p < 0.0001$. See also Fig. EV4. Source data are available online for this figure.

## LB-100-driven alternative splicing results in the formation of neopeptides presented by cancer cells

Recent data indicate that splicing inhibition downstream of SF3B1 and RBM39 results in the generation of neoantigens and the extension of lifespan in tumor-bearing mice (Lu et al, 2021). Likewise, LB-100 has been shown to sensitize different cancer models to immune checkpoint inhibitors with divergent underlying mechanisms proposed (Ho et al, 2018; Maggio et al, 2020; Stanford and Bottini, 2023). These observations led to the question of

whether the ASEs induced by LB-100 in colorectal adenocarcinoma cells could also be a source of neoantigens, providing further mechanistic support to the combination with immune checkpoint inhibitors. Interestingly, we observed that mRNAs that were alternatively spliced-in response to LB-100 treatment strongly overlapped with reported immunopeptide-generating alternatively spliced transcripts in spliceosome-inhibited cells (Fig. 5A). On average, 50% of the alternatively spliced mRNAs from all ASE types were predicted to generate cancer neoantigens (Fig. EV5A), consistent with previous findings (Lu et al, 2021). To investigate

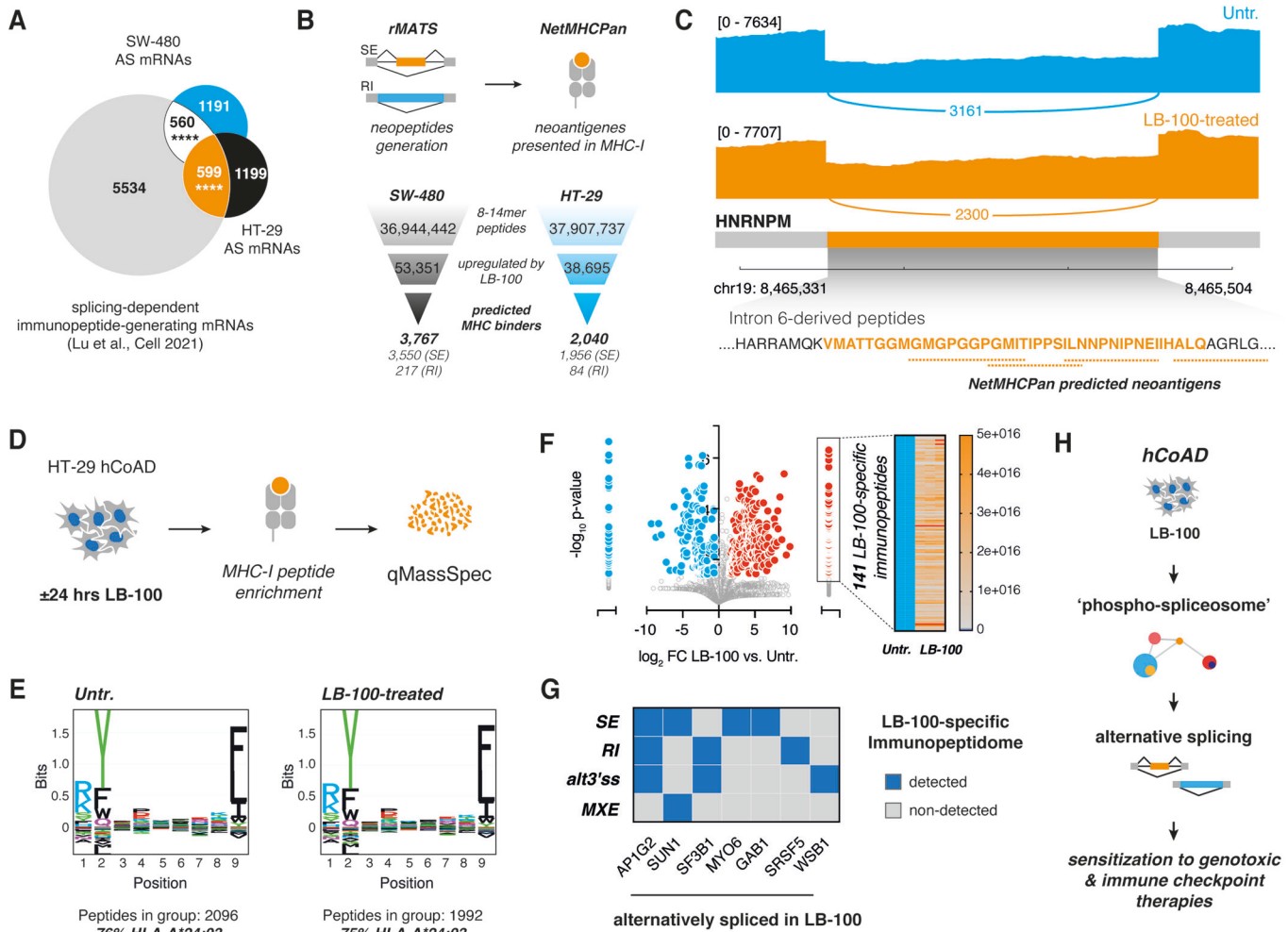

**Figure 5. LB-100-triggered alternative splicing changes increase neoepitope production.**

(A) Venn diagram shows an overlap between alternatively spliced mRNAs in SW-480 and HT-29 colon adenocarcinomas and previously reported neoantigen-producing mRNAs (Lu et al, 2021). (B) In silico prediction of MHC-I binders derived from alternative splicing events triggered by LB-100 in SW-480 and HT-29 human colorectal adenocarcinomas. (C) Alternative splicing event within HNRNPM gene and representative Sashimi plots showing differential splicing in untreated and LB-100-treated SW-480 cells. Intron 6-derived peptides predicted to bind MHC-I are underlined. (D) Experimental setup employed to capture immunopeptides presented in MHC-I context in response to PP2A/PP5 inhibitor LB-100 in human colorectal adenocarcinoma (hCoAD) cell line HT-29. (E) Sequence logo for 9-mers identified from the MHC-I immunopurification in untreated controls (Untr.) and LB-100-treated HT-29 colorectal adenocarcinoma cell line. (F) The volcano plot shows changes in the MHC-I presentation of individual peptides in response to LB-100 stimulation compared to untreated controls. Heat map shows levels of detection for 141 significant (*p* value <0.05, *t*-test) peptides detected only in LB-100-treated cells compared to untreated controls (Untr.) in *n* = 3 biological replicates. (G) The graph depicts the type of detected alternative splicing events in seven transcripts being a source for peptides detected only in LB-100-treated cells. (H) Working model. LB-100-mediated phosphorylation changes in spliceosome drive differential splicing patterns in human colorectal carcinoma. Predominant exon skipping and intron retention events affect DNA damage responses and neoantigen formation, with possible implications for therapy response. Data information: In (A), statistical significance was calculated with a hypergeometric test, ****p < 0.0001. See also Fig. EV5 and Datasets EV5, EV6.

further the potential of LB-100 for generation of the immune-proficient neopeptides, we defined predicted binding affinities of retained intron and skipped exon-derived neopeptides using the artificial intelligence-assisted tool NetMHCpan (Reynisson et al, 2020) (Fig. 5B). This analysis revealed multiple potential MHC-I-bound peptides formed downstream of LB-100-dependent splicing alterations (Fig. 5B), including previously reported (Lu et al, 2021) neoantigen-generating alternatively spliced HNRNPM mRNA (Fig. 5C).

Finally, to confirm that LB-100 treatment does generate neopeptides that are presented in the context of MHC-I, we

augmented our in silico analyses with mass spectrometry-based immunopeptidomics of colon cancer cells treated with LB-100 for 24 h (Fig. 5D). Given that SW-480 cells showed intrinsically low expression of MHC-I, we employed HT-29 cells in these analyses. After enrichment for MHC-I-bound peptides from control and LB-100-treated cells, we performed liquid chromatography-tandem mass spectrometry (LC-MS/MS) to determine the repertoire of immunopeptides resulting from LB-100 treatment (Fig. 5D). Most of the identified peptides in control and LB-100-treated cells were presented in the context of HLA-A*24:03, with expected sequence preferences at anchor residues (Figs. 5E and EV5B). Notably, our

data show an increased number of peptides presented in the context of MHC-I in LB-100-treated cells compared to untreated controls (Figs. 5F and EV5C). Furthermore, we identified 141 peptides derived from 138 proteins that could be recovered only from the LB-100-treated cells (Figs. 5F and EV5C). The intersection of these 138 source proteins with alternatively spliced targets of LB-100 revealed seven prime candidates undergoing multiple splicing alterations (Fig. 5G).

Altogether, these data show that the rewiring of the spliceosome phosphorylation induced by LB-100 results in ASEs that are translated into drug-induced neopeptides effectively presented by the MHC-I of colon cancer cells. Moreover, these observations place LB-100 as a prospective compound for splicing-targeting therapy aiming to enhance the efficacy of immunotherapy approaches for cancer treatment (Fig. 5H).

Collectively, our current work identifies a major role for LB-100 in the modulation of RNA splicing. Our data indicate that inhibition of PP2A/PP5 with this small molecule inhibitor leads to major changes in the phosphorylation of splicing proteins, with both hyper- as well as hypophosphorylation events. That results in aberrant RNA splicing events, particularly skipped exons and retained introns. We propose that while the upregulated phospho-sites can be direct targets of PP2A/PP5, the hypophosphorylation of specific sites is likely an effect of secondary events evoked by PP2A/PP5 inhibition. The impact of LB-100 on the phosphorylation of splicing factors by phosphatase inhibition is supported by other phosphoproteomic analyses (Hoermann et al, 2020; Kauko et al, 2020). However, the functional impact of individual differential phosphorylation events merits further detailed investigation.

Attempts to disturb the splicing patterns of cancer cells to generate neoantigens that could boost immunotherapies have gained traction recently. Our findings indicate that LB-100 induces adverse splicing events in colorectal cancer cells, which overlap with events previously reported to generate neoantigens recognized by cytotoxic T cells. For instance, studies involving the molecular glue indisulam, which causes selective degradation of the splicing factor RBM39 (Han et al, 2017), demonstrated that the resulting mis-splicing events create neoantigens, thereby enhancing the efficacy of immune checkpoint blockade (Lu et al, 2021). Interestingly, we found that two major phosphorylation sites on RMB39 are affected by LB-100, corroborated further by the observation that the adverse splicing events induced by LB-100 overlap with the events previously reported to generate neoantigens recognized by cytotoxic T cells after indisulam treatment (Lu et al, 2021). In fact, multiple studies in a variety of cancer models have demonstrated that LB-100 is synergistic with immune checkpoint blockade (Ho et al, 2018; Maggio et al, 2020; Yen et al, 2021). Furthermore, patients carrying a mutation in the PP2A scaffold protein PP2R1A have been reported to exhibit a superior response to immune checkpoint blockade therapy (Yen et al, 2021). It is uncertain at this point whether the observed synergy between LB-100 and immune checkpoint blockade can be primarily attributed to the generation of neoantigens as a result of adverse splicing events, as LB-100 also has other immune modulatory effects. For example, it was shown that ablation of PP2A in regulatory T cells (Tregs) inhibits their immunosuppressive activity and causes auto-immunity (Apostolidis et al, 2016). Moreover, LB-100 in glioma activates the cGAS-STING pathway, leading to the activation of interferon signaling and an increase in CD8+ killer T-cell

proliferation (Ho et al, 2023; Mondal et al, 2023). The role of PP2A inhibition in increased T-cell proliferation has also been observed by others (Zhou et al, 2014).

Although the efficacy of combining LB-100 and immune checkpoint blockade has been demonstrated *in vivo*, our data provide a more mechanistic explanation for this effect, by showing LB-100-induced neopeptides derived from mis-spliced transcripts presented by MHC-I. Together, these data provide a strong rationale for testing the combination of LB-100 and immune checkpoint blockade in the clinic. A first trial testing this was started recently (https://classic.clinicaltrials.gov/ct2/show/NCT04560972).

## Methods

### Cell culture

Human colorectal adenocarcinoma cells SW-480 and HT-29 were purchased from ATCC and maintained in, respectively, Leibovitz's L-15 Medium and McCoy's 5 A medium supplemented with 10% fetal bovine serum (FBS) (Thermo Fisher Scientific) and 1% penicillin/streptomycin (Thermo Fisher Scientific). All cells were grown at 37 °C with 5% $CO_2$ and routinely tested for mycoplasma infection (Universal Mycoplasma Detection Kit, ATCC).

### Synergy matrices

All synergy matrices were performed in triplicate, using black-walled 384-well plates (Greiner #781091). Cells were plated at 10–20% density and incubated overnight for attachment. Drugs were then added to the cells using the Tecan D300e digital dispenser. Z-score above 10 was considered to indicate a positive synergy of action between two drugs.

### Immunoblotting analysis of protein expression

After harvesting the cells at the indicated time points of drug treatment, cells were lysed with RIPA buffer (25 mM Tris-HCl, pH 7.6, 150 mM NaCl, 1% NP-40, 1% sodium deoxycholate, 0.1% SDS) containing Complete Protease Inhibitor cocktail (Roche) and phosphatase inhibitor cocktails II and III (Sigma). Samples were cleared by centrifugation at $15,000 \times g$ for 15 min at 4 °C and supernatant was collected. Protein concentration was calculated with Bicinchoninic Acid (BCA) assay (Pierce BCA, Thermo Scientific), to ensure equal sample loading. DTT-denatured protein samples were then loaded in a 4–12% gradient polyacrylamide gel and resolved by SDS-PAGE for approximately 40 min at 180 V. Proteins were transferred from the gel to a nitrocellulose membrane at 350 mA for 90–120 min. After the transfer, membranes were incubated in a blocking solution (1% bovine serum albumin—BSA, or 1% non-fat dry milk in TBS with 0.1% Tween-20 (TBS-T). Subsequently, membranes were probed with the primary antibodies in TBS-T with 5% BSA overnight at 4 °C. After 3 × 10 min washes with TBS-T, membranes were incubated for 1 h at room temperature with the HRP-conjugated secondary antibody in blocking solution. Membranes were again washed three times for 5 min in TBS-T and visualized with a chemiluminescence substrate (ECL, Bio-Rad) using the ChemiDoc-Touch (Bio-Rad). Antibodies

were purchased from Cell Signaling Technologies (ATM, CAT#2873 and Chk2, CAT#2662, or from Sigma-Aldrich (a-Tubulin, CAT#T9026).

## RNA-sequencing

Transcriptome-wide splicing analysis was performed using SW-480 and HT-29 cell lines. These cells were treated with LB-100 (4 μM) for 8 h, with untreated cells serving as a reference control for the analysis. Total RNA from the cells was isolated using the RNeasy Mini Kit (Qiagen), which included an on-column DNase digestion (Qiagen), according to the manufacturer's instructions. The quality and quantity of isolated RNA were assessed using the RNA 6000 Nano LabChip Bioanalyzer (Agilent Technologies). Sequencing libraries were constructed from total RNA using the TruSeq Stranded mRNA library kit (Illumina). The quality of the resulting mRNA libraries was evaluated using a 2100 Bioanalyzer instrument, following the manufacturer's protocol specified for the Agilent DNA 7500 kit (Agilent Technologies). After dilution to 10 nM and equimolar pooling into multiplex sequencing pools, paired-end sequencing was performed on the NovaSeq 6000 (Illumina) sequencing instrument.

## Proteomics LC-MS/MS mass spectrometry

Frozen LB-100- or control-treated SW-480 cell pellets were lysed in Guanidine (GuHCl) lysis buffer as described (Jersie-Christensen et al, 2016). Lysate protein concentrations were determined with a Pierce Coomassie (Bradford) Assay Kit (Thermo Scientific) according to the manufacturer's instructions. Aliquots corresponding to 1100 μg of protein were diluted to 2 M GuHCl and digested twice (4 h and overnight) with trypsin (Sigma-Aldrich) at 37 °C, enzyme/substrate ratio 1:75. Digestion was quenched by the addition of TFA (final concentration 1%), after which the peptides were desalted on a Sep-Pak C18 cartridge (Waters, MA, USA). From the eluates, 50-μg-aliquots were collected for proteome analysis, the remainder being reserved for phosphoproteome analysis. Samples were vacuum-dried and stored at −80 °C until LC-MS/MS analysis or phosphopeptide enrichment. Phosphopeptides were enriched using the High Select Phosphopeptide Enrichment Kits (Pierce) according to the manufacturer's instructions, after which eluates were vacuum-dried until LC-MS/MS.

Single-shot LC-MS/MS of proteome samples was performed by nanoLC-MS/MS on an Orbitrap Exploris 480 mass spectrometer (Thermo Scientific) connected to a Proxeon nLC1200 system. Peptides were directly loaded onto the analytical column (ReproSil-Pur 120 C18-AQ, 2.4 μm, 75 μm × 500 mm, packed in-house) and eluted in a 90-min gradient containing a linear increase from 6 to 30% solvent B (solvent A was 0.1% formic acid/water and solvent B was 0.1% formic acid/80% acetonitrile). The Exploris 480 was run in data-independent acquisition (DIA) mode, with full MS resolution set to 120,000 at m/z 200, MS1 mass range was set from 350–1400, normalized AGC target was 300% and maximum IT was 45 ms. DIA was performed on precursors from 400–1000 in 48 windows of 13.5 Da with an overlap of 1 Da. Resolution was set to 30,000 and normalized CE was 27.

For single-shot LC-MS/MS of phosphoproteome samples, a 135-min gradient containing a 114-min linear increase from 7 to 30%

solvent B was used. The Exploris 480 was run in data-dependent acquisition (DDA) mode with full MS resolution set to 120,000 at m/z 200 and a cycle time of 2 s. MS1 mass range was set from 375 to 1500, the normalized AGC target was 300%, and maximum IT mode was set to "Auto". Exclusion duration was set to 30 s; MS2 spectra were acquired at 30,000 resolution with data-dependent mode set to "cycle time". Precursors were HCD-fragmented with a normalized collision energy of 27 when their charge states were 2–6; the MS2 isolation window was 1.2 m/z, the normalized AGC target was set to "standard" and the maximum injection time mode was set to "auto".

## Immunoproteome prediction

For in silico prediction of MHC-I binders, we build libraries consisting of sequences of transcripts with decreased skipped exons (SE) and increased intron retention (RI) events upon LB-100 treatment, reasoning that these will be productive in terms of neopeptide formation. For background, annotated coding sequences were extracted. Afterwards, sequences were computationally translated into proteins and digested into unique 8–14mer peptides. Each of the input nucleotide sequences was translated with the following assumptions: (1) only the annotated, canonical START sites were used; (2) no stop codon readthrough or internal translation initiation occurred. We predicted binding affinities of the resulting 8–14mers to the HLA-A02:01 MHC-I haplotype by NetMHCpan v4.1 (Reynisson et al, 2020).

## MHC-I immunoprecipitation and mass spectrometry analysis of neoepitopes

For immunoprecipitation of HLA-peptides, $10^8$ cells per replicate were seeded and treated or not with LB-100 for 24 h. After treatment, cells were washed with PBS and harvested by trypsinization. Cells were then collected and centrifuged (5 min at 1500 rpm) and then washed with PBS. The dry pellet was snap-frozen in liquid nitrogen. Then, the cell pellet was lysed as described previously (Chong et al, 2020; Pataskar et al, 2022). W6/32 antibody cross-linked to protein-A sepharose 4B beads was used for the immunoaffinity purification.

For mass spectrometry detection of neopeptides, after vacuum concentration of the IP eluates, peptides were analyzed by LC-MS/MS on an Orbitrap Exploris 480 Mass spectrometer connected to an Evosep One LC system (Evosep Biotechnology, Odense, Denmark). Before LC separation, peptides were reconstituted in 0.1% formic acid, and 50% of the sample was loaded on Evotip Pure™ (Evosep) tips. Peptides were then eluted and separated using the pre-programmed "Extended Method" (88 min gradient) on an EV1137 (Evosep) column with an EV1086 (Evosep) emitter. Nanospray was achieved using the Easy-Spray NG Ion Source (Thermo Scientific) with a liquid junction set-up at 1.9 kV.

Prior to LC separation with the nLC1200, peptides were reconstituted in 2% formic acid, after which 50% of the sample was directly loaded onto the analytical column (ReproSil-Pur 120 C18-AQ, 2.4 μm, 75 μm × 500 mm column, packed in-house in fritted Empty Self Pack NanoLC column tubes with integrated emitter tip (CoAnn Technologies LLC, WA, United States). Peptides were eluted in a 110-min gradient containing a linear increase from 6 to 30% solvent B (solvent A was 0.1% formic acid/

water and solvent B was 0.1% formic acid/80% acetonitrile) followed by washout at 90% solvent B. Nanospray was achieved using the Nanospray Flex™ Ion source (Thermo Scientific) with a liquid junction set-up at 2.0 kV.

On the Exploris 480, data-dependent acquisition was performed as follows. Full scan MS was acquired at resolution 60,000 with MS1 mass range 350–1700 m/z, normalized AGC target was set to 100%, and maximum injection time was 50 ms. Dynamic exclusion was set to 10 s, and MS2 spectra were acquired at 15,000 resolution. The top ten precursors per cycle were HCD-fragmented when their charge states were 2–4, whereas the top five precursors per cycle were subjected to HCD fragmentation if they were singly charged. The MS2 isolation window was 1.1 m/z, the normalized collision energy was 30, the normalized AGC target was set to 50%, and the maximum injection time was 100 ms.

The immunopeptidomics data have been deposited to ProteomeXchange along with the (phosho)proteome data.

## RNA-seq and splicing analysis

Paired-end sequencing was performed using 101 cycles for Read 1, 19 cycles for Read i7, 10 cycles for Read i5, and 101 cycles for Read 2, using the NovaSeq 6000 SP Reagent Kit v1.5 (200 cycles) (20040719, Illumina). Sequencing data was demultiplexed into FastQ files using BCLConvert version 3.9.3 (Illumina). The paired-end reads were trimmed for adapter sequences using SeqPurge version 2019_09 (Sturm et al, 2016) and aligned to the human hg38 reference genome using HiSat2 (Kim et al, 2019) version 2.1.0 with the pre-built grch38_snp_tran reference. Gene counts were generated using HTseq-count (Anders et al, 2015) and Homo_sapiens.GRCh38.102.gtf. Splicing events were determined using rMATS version 4.1.2 (Shen et al, 2014) using the Ensembl release 100 transcript GTF (Yates et al, 2019). An ASE was considered significant based on the cut-off for FDR <0.05 and |ΔPSI| >0.1 for the individual condition supporting the existence of a transcript with ASE. Analysis of sequence features associated with identified ASEs was performed with MATT version 1.2.1 (Gohr and Irimia, 2019), comparing 410 LB-100-induced (UP) and 641 LB-100-repressed (DOWN) skipped exons or 87 (UP) and 105 (DOWN) retained introns to the background (NO). The enrichment for RNA binding protein sites within LB-100-regulated exons was calculated using the CISBP-RNA binding motif RNA-maps (Ray et al, 2013) using default settings. The following groups were used for comparison: exons with increased (UP) and decreased (DOWN) inclusion upon LB-100 treatment of SW-480 cells, and reference group (NO; FDR >0.05 and |ΔPSI| >0.1). Gene Ontology (GO) analysis was performed using Enrichr.

## Proteomics data analysis

Proteome data were analyzed with DIA-NN (version 1.8) (Demichev et al, 2020) without a spectral library and with the "Deep learning" option enabled. The SwissProt Human database (20,395 entries, release 2021_04) was added for the library-free search; the quantification strategy was set to "Robust LC (high accuracy)", and the MBR option was enabled. All other settings were kept at the default values. The protein groups report from DIA-NN was used for downstream analysis in Perseus (version:

1.6.15.0) (Tyanova et al, 2016). Values were log2-transformed, after which proteins were filtered for at least 75% valid value presence in at least one sample group. Missing values were replaced by imputation based on a normal distribution using a width of 0.3 and a minimal downshift of 2.4. Differential protein abundances were determined using a student's t-test (minimal threshold: FDR: 5% and S0: 0.35). Phosphoproteome data were analyzed using MaxQuant (version 2.0.3.0) (Cox et al, 2014) using standard settings for label-free quantitation (LFQ). MS/MS data were searched against the same sequence database as mentioned above, complemented with a list of common contaminants, and concatenated with the reversed version of all sequences. The maximum allowed mass tolerance was 4.5 ppm in the main search and 0.5 Da for fragment ion masses. False discovery rates for peptide and protein identification were set to 1%. Trypsin/P was chosen as cleavage specificity allowing two missed cleavages. Carbamidomethylation was set as a fixed modification, with methionine oxidation and Phospho(STY) as variable modifications. Phospho-site LFQ intensities were extracted from the Phospho(STY)sites.txt file and processed in Perseus as described above and filtered for reverse sequences and potential contaminants, as well as for site probability ≥0.75. Additionally, sites were filtered for at least 100% valid value presence in at least one sample group. Values were normalized by median subtraction and missing values were replaced by imputation based on a normal distribution (width: 0.3 and downshift: 1.8). Phosphosites with differential abundance were determined using a Student t-test (minimal threshold: FDR: 5% and S0: 0.1). All mass spectrometry proteomic data generated in this study have been deposited to the ProteomeXchange Consortium via the PRIDE (Perez-Riverol et al, 2022) partner repository.

## Quantification and statistical analysis

Data were presented as mean, ±SD, or SEM, unless otherwise stated. An indicated number of independent biological replicates have been performed for each experiment. Statistical tests are used, and specific p values are indicated in the figure legends.

# Data availability

RNA-seq data have been deposited to the GEO database (https://www.ncbi.nlm.nih.gov/geo/) under the following accession number: GSE236625. Results of RNA-seq for time-resolved gene expression changes in response to LB-100 can be accessed at https://zenodo.org/records/10640576. Proteomics data have been deposited to the ProteomeXchange Consortium via the PRIDE (Perez-Riverol et al, 2022) partner repository with the identifier PXD043658.

# Peer review information

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

## Acknowledgements

We thank members of the Bernards and Cieśla laboratories for helpful discussion and thoughtful feedback. We thank the Genomics core facility of the Netherlands Cancer Institute for RNA-seq experiments and the Protein facility for carrying out the proteomics experiments. We acknowledge Michał Krzysztoń for critical input on data analysis. This work was supported by an institutional grant from the Dutch Cancer Society and of the Dutch Ministry of Health, Welfare and Sport by the Oncode Institute (RB) and by a research grant from Lixte Biotechnology (RB). MC is a beneficiary of Sonata Bis (UMO-2022/46/E/NZ3/00141) and OPUS (UMO-2021/43/B/NZ3/01177) grants from the National Science Center in Poland and Installation Grant from European Molecular Biology Organization. OBB received support from the X-omics Initiative (Project 184.034.019), part of the NWO National Roadmap for Large-Scale Research Infrastructures. We thank the Reviewers of this manuscript for providing valuable feedback during revision. We thank EMBO for subsidizing publication of this article in frame of Installation Grant for MC.

## Author contributions

**Matheus H Dias**: Formal analysis; Investigation; Methodology; Writing—original draft; Writing—review and editing. **Vladyslava Liudkovska**: Data curation; Software; Formal analysis; Methodology; Writing—review and editing. **Jasmine Montenegro Navarro**: Formal analysis; Investigation. **Lisanne Giebel**: Investigation. **Julien Champagne**: Resources; Investigation. **Chrysa Papagianni**: Investigation. **Onno B Bleijerveld**: Resources; Formal analysis; Investigation; Methodology. **Arno Velds**: Resources; Data curation; Software; Writing—review and editing. **Reuven Agami**: Resources; Writing—review and editing. **René Bernards**: Conceptualization; Supervision; Funding acquisition; Investigation; Writing—original draft; Project administration; Writing—review and editing. **Maciej Cieśla**: Conceptualization; Formal analysis; Supervision; Writing—original draft; Project administration; Writing—review and editing.

## Disclosure and competing interests statement

RB received research funding from Lixte Biotechnology Holdings, the company that manufactures LB-100. Rene Bernards is also a member of the board of directors of Lixte Biotechnology Holdings Ltd. MHD is a shareholder of Lixte Biotechnology. RA is a member of the Advisory Editorial Board of EMBO Reports, which has no bearing on the editorial consideration of this article for publication. The remaining authors declare no competing interests.

# Expanded View Figures

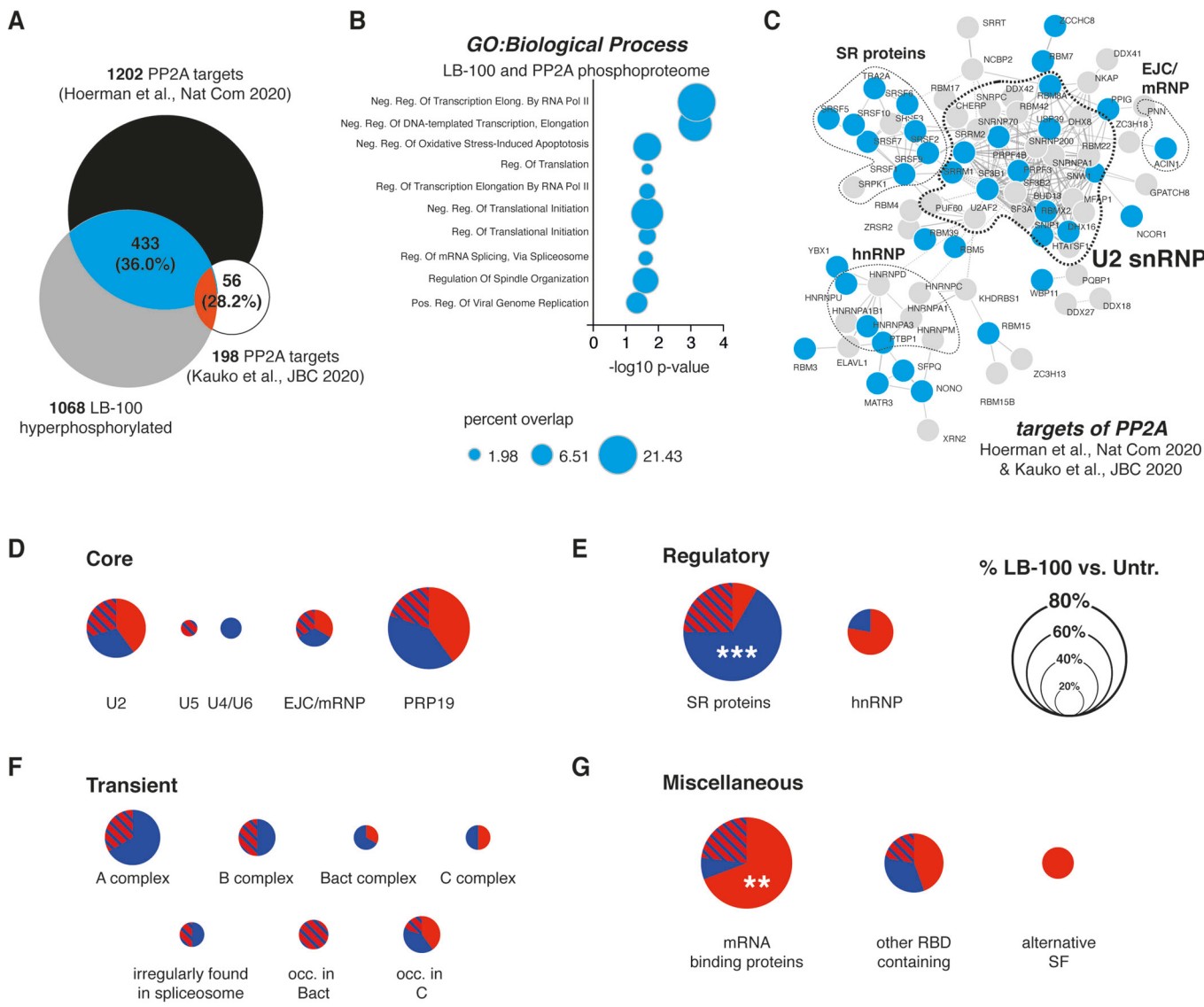

**Figure EV1.   Specific splicing nodes are differentially phosphorylated in response to LB-100 treatment in colorectal adenocarcinoma.**

(A) Venn diagram shows an overlap between differentially hyperphosphorylated proteins in LB-100-treated SW-480 colon adenocarcinomas and previously reported targets of PP2A (Hoermann et al, 2020; Kauko et al, 2020). (B) The bar graph shows Gene Ontologies enriched for 34 differentially phosphorylated proteins from LB-100-treated SW-480 human colorectal carcinoma that are previously reported PP2A targets (Hoermann et al, 2020; Kauko et al, 2020). *p* values were calculated with Fisher's exact test. (C) Interaction maps of spliceosome components hyper- and hypophosphorylated in response to 12 h of LB-100 stimulation in SW-480 cells (gray) overlaid with previously reported targets of PP2A (Hoermann et al, 2020; Kauko et al, 2020) (blue). Connecting lines show interactions from STRINGdb. (D–G) Pie charts show the distribution of proteins with significantly changed (FDR <0.05 and |Log$_2$Fold Change| >0.58) phosphorylation status within different functional groups of splicing factors. RBD RNA binding domain, SF splicing factor. Data information: In (D–G), statistical significance was calculated with hypergeometric test vs. background, ***$p$ < 0.001; **$p$ < 0.01.

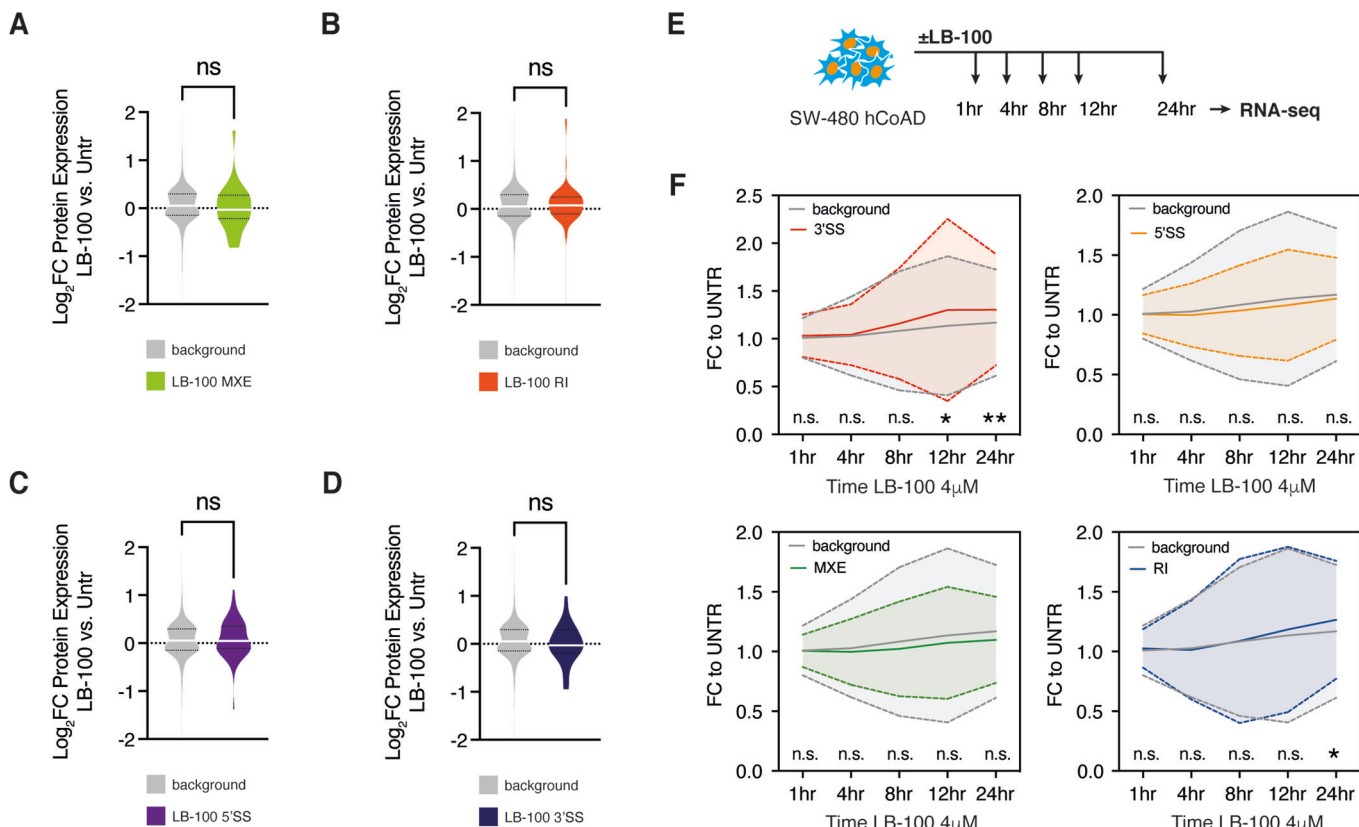

**Figure EV2. PP2A inhibition-dependent alternative splicing affects protein but not transcript levels of alternatively spliced mRNAs.**

(A–D) Violin plots show no effect on levels of proteins resulting from alternatively spliced transcripts in MXE (A), RI (B), alt5′ss (C), and alt3′ss (D) in 4 μM LB-100-treated SW-480 cells compared to control in *n* = 4 biological replicates. (E) Schematic illustrates an experimental approach to determine time-resolved transcriptomic changes of alternatively spliced mRNAs in response to 4 μM LB-100 stimulation. (F) Graphs show time-resolved changes in alternatively spliced transcripts and mRNAs from RI, MXE, alt5′ss, and alt3′ss groups in response to LB-100 stimulation. Data information: In (F), statistical significance was calculated with one-way ANOVA, **$p < 0.01$; *$p < 0.05$; n.s. not significant. Source data are available online for this figure.

 

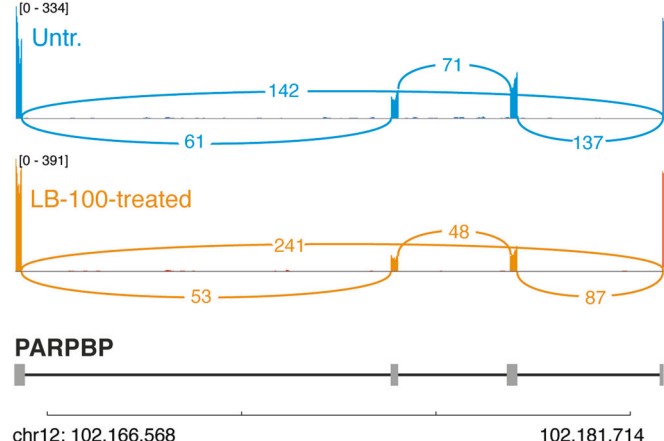

**Figure EV3.  Alternative splicing events in response to LB-100 impact the fate of DNA damage-response regulators.**

An alternative splicing event within PARPBP gene and representative Sashimi plots showing differential splicing in untreated and LB-100-treated SW-480 cells.

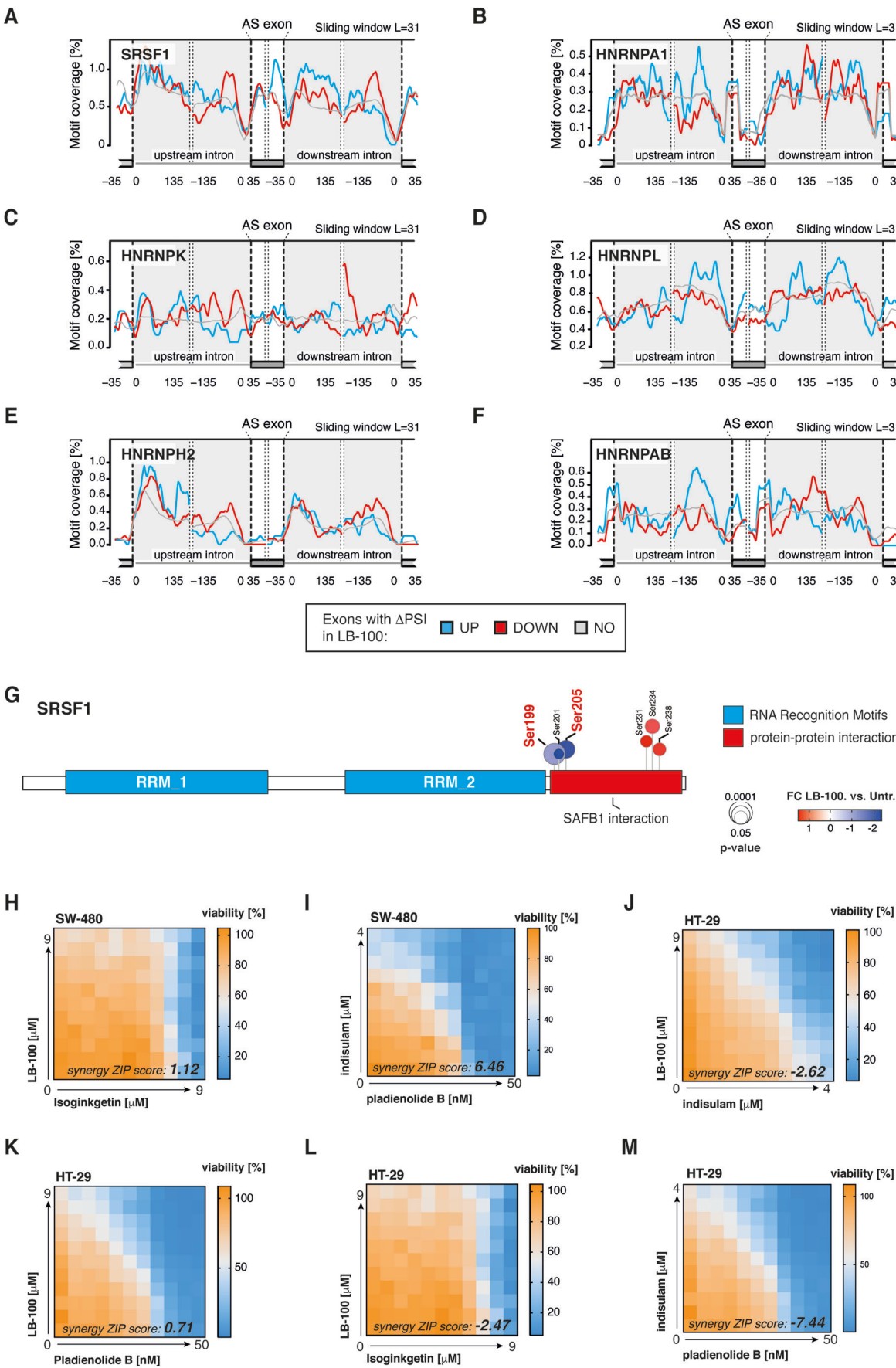

**Figure EV4. Impact of LB-100-driven phosphorylation changes on the function of spliceosome components.**

(A–F) Motif RNA map shows SRSF1 (**A**), HNRNPA1 (**B**), HNRNPK (**C**), HNRNPL (**D**), HNRNPH2 (**E**), and HNRNPAB (**F**) proteins binding in the proximity of excluded (DOWN), included (UP), or not affected (NO) exons in LB-100-treated as compared to untreated SW-480 colorectal adenocarcinoma cells. (**G**) Location of mis-phosphorylated sites across SRSF1 protein. Functionally relevant protein regions according to UniProt are shown. Significantly changing phosphosites in response to LB-100 are highlighted. (**H,I**) Synergy matrices show a lack of cooperativity between LB-100 and isoginkgetin (**H**) or indisulam and pladienolide B (**I**) in the SW-480 human adenocarcinoma cell line. Representative heat map for $n = 3$ biological replicates. (**J–M**) Synergy matrices show a lack of cooperativity between LB-100 and indisulam (**J**), pladienolide B (**K**), isoginkgetin (**L**), or indisulam and pladienolide B (**M**) in HT-29 human adenocarcinoma cell line. Representative heat map for $n = 3$ biological replicates. Source data are available online for this figure.

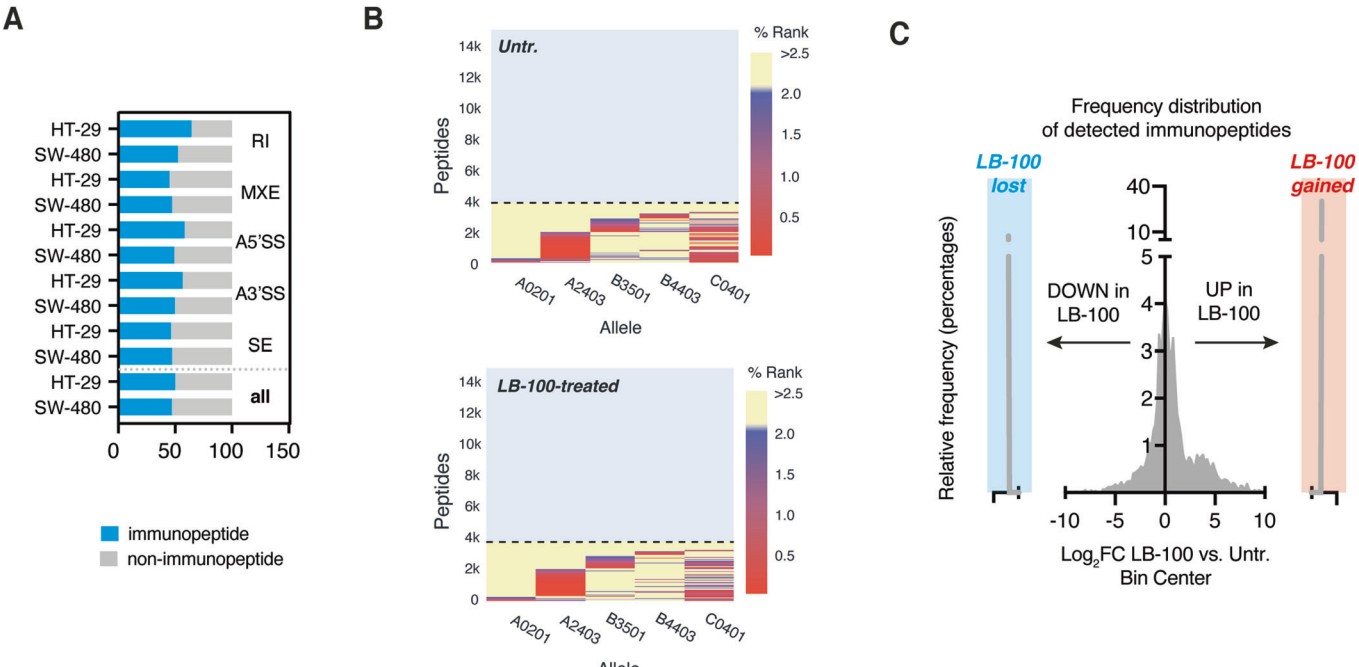

**Figure EV5. LB-100 determines neoantigen formation presented within MHC-I.**

(A) The bar graph shows an equal contribution of all the splicing events toward the potential formation of tumor neoantigens reported previously (Lu et al, 2021). (B) Heat maps show HLA haplotype representation in untreated controls (Untr.). and LB-100-stimulated HT-29 human colorectal carcinoma cells from $n = 3$ biological replicates. (C) The plot shows the cumulative distribution of $\log_2$ fold change in peptides presentation between LB-100-stimulated and untreated control (Untr.) HT-29 cells from $n = 3$ biological replicates. Highlighted in red and blue are peptides detected exclusively in LB-100 and control cells, respectively.

