## [Peer Review File · EMBO Reports]

The phosphatase inhibitor LB-100 creates neoantigens in colon cancer cells through perturbation of mRNA splicing

Matheus Dias, Vladyslava Liudkovska, Jasmine Montenegro Navarro, Lisanne Giebel, Julien Champagne, Chrysa Papagianni, Onno B. Bleijerveld, Arno Velds, Reuven Agami, Rene Bernards, and Maciej Ciesla

Corresponding author(s): Maciej Ciesla (m.ciesla@imol.institute) , Rene Bernards (r.bernards@nki.nl)

Review Timeline:

Submission Date:	18th Aug 23
Editorial Decision:	21st Sep 23
Revision Received:	12th Feb 24
Editorial Decision:	6th Mar 24
Revision Received:	15th Mar 24
Accepted:	20th Mar 24

Editor: Achim Breiling

Transaction Report:

Dear Dr. Ciesla,

Thank you for the submission of your manuscript to EMBO reports. I have already forwarded to you the reports I received from the three referees that I asked to evaluate your study. Please find them again below. As you know, the referees think that these findings are of interest. However, they have several comments, concerns, and suggestions to improve the manuscript.

Given the constructive referee comments and after going through your preliminary point-by-points response (revision plan), I decided to proceed with the manuscript. Please address the referee concerns as indicated in your letter in a revised manuscript and/or in a detailed final point-by-point response.

Acceptance of your manuscript will depend on a positive outcome of a second round of review. It is EMBO reports policy to allow a single round of revision only and acceptance of the manuscript will therefore depend on the completeness of your responses included in the next, final version of the manuscript.

- 1) a .docx formatted version of the final manuscript text (including legends for main figures, EV figures and tables), but without the figures included. Figure legends should be compiled at the end of the manuscript text.
- 2) individual production quality figure files as .eps, .tif, .jpg (one file per figure), of main figures (up to 8) and EV figures (up to 5). Please upload these as separate, individual files upon re-submission.

- 4) a complete author checklist, which you can download from our author guidelines

(<https://www.embopress.org/page/journal/14693178/authorguide>). Please insert page numbers in the checklist to indicate where the requested information can be found in the manuscript. The completed author checklist will also be part of the RPF.

- 5) that primary datasets produced in this study (e.g. RNA-seq, ChIP-seq, structural and array data) are deposited in an appropriate public database. If no primary datasets have been deposited, please also state this in a dedicated section (e.g. 'No

primary datasets have been generated and deposited'), see below.

The accession numbers and database should be listed in a formal "Data Availability" section (placed after Materials & Methods) that follows the model below. This is now mandatory (like the COI statement). Please note that the Data Availability Section is restricted to new primary data that are part of this study. This section is mandatory. As indicated above, if no primary datasets have been deposited, please state this in this section

Data availability

8) Regarding data quantification and statistics, please make sure that the number "n" for how many independent experiments were performed, their nature (biological versus technical replicates), the bars and error bars (e.g. SEM, SD) and the test used to calculate p-values is indicated in the respective figure legends (also for potential EV figures and all those in the final Appendix). Please also check that all the p-values are explained in the legend, and that these fit to those shown in the figure. Please provide statistical testing where applicable. Please avoid the phrase 'independent experiment', but clearly state if these were biological or technical replicates. Please also indicate (e.g. with n.s.) if testing was performed, but the differences are not significant. In case n=2, please show the data as separate datapoints without error bars and statistics. See also: <http://www.embopress.org/page/journal/14693178/authorguide#statisticalanalysis>

9) If you present microscopic images, please add scale bars of similar style and thickness, using clearly visible black or white bars (depending on the background). Please place these in the lower right corner of the images themselves. Please do not write on or near the bars in the image but define the size in the respective figure legend.

10) Please also note our reference format:

12) We now use CRedit to specify the contributions of each author in the journal submission system. CRedit replaces the author contribution section. Please use the free text box to provide more detailed descriptions and do not provide your final manuscript text file with an author contributions section. See also our guide to authors: <https://www.embopress.org/page/journal/14693178/authorguide#authorshipguidelines>

13) We would encourage you to use 'Structured Methods', our new Materials and Methods format. According to this format, the Materials and Methods section should include a Reagents and Tools Table (listing key reagents, experimental models, software

and relevant equipment and including their sources and relevant identifiers) followed by a Methods and Protocols section in which we encourage the authors to describe their methods using a step-by-step protocol format with bullet points, to facilitate the adoption of the methodologies across labs. More information on how to adhere to this format as well as downloadable templates (.doc or .xls) for the Reagents and Tools Table can be found in our author guidelines (section 'Structured Methods'):

14) Please add up to 5 keywords to the title page and order the manuscript sections like this, using these names:

Title page - Abstract - Keywords - Introduction - Results - Discussion - Materials and Methods - Data availability section - Acknowledgements - Disclosure and Competing Interests Statement - References - Figure legends - Expanded View Figure legends

I look forward to seeing a revised version of your manuscript when it is ready. Please let me know if you have questions or comments regarding the revision.

Yours sincerely,

Referee #1:

The manuscript by Dias and co-workers analyzed differentially expressed proteomes and phosphoproteomes in response to 12-hour treatment with broad-specificity serine threonine phosphatase inhibitor LB100. LB100 and its derivative LB102 are in clinical development as cancer therapies, and they synergize with genotoxic therapies and induce anticancer immune responses. The work by Dias et al., is technically sound and the results interesting, but the authors totally neglect existing literature that none of the cantharidin derivatives, including LB100, cannot be considered as selective PP2A inhibitors (PMID: 27002182, 17200551 etc.). In fact, LB100 was recently shown to be equally efficient inhibitor of another serine threonine phosphatase PP5 than PP2A (PMID: 30679389). Further, the anti-tumor effects of LB100 that the authors call surprising, are not surprising as also other broad-specificity serine threonine phosphatase inhibitors such as okadaic acid are cytotoxic to cancer cells, and because the target of LB100, PP5, is oncogenic (PMID: 37527661, 29805615). Therefore, the current manuscript is severely misleading, and there is no basis for the main conclusions that the observed effects by LB100 would be mediated by selective inhibition of PP2A. In fact, none of the LB100 papers published so far have been able to convincingly demonstrate PP2A selectiveness of the reported effects.

There are two alternative strategies to solve this major issue:

1. Rewriting the entire manuscript so that all mentions related to PP2A are removed, and the effects are described solely as pharmacological responses to LB100. This must be supplemented by careful discussion of the existing literature about lack of the non-selectivity of LB100 against serine threonine phosphatases. This strategy would include NOT starting the introduction with PP2A, but carefully introducing the evidence that LB equally well inhibits PP5 and PP2A and the expected cancer selective effects of inhibitions of these phosphatases.
2. Providing solid validation data demonstrating that the major conclusions can be recapitulated by direct inhibition of PP2A by for example siRNA or CRISPR/Cas9. Parallel to that, authors should compare the identified splicing targets to known PP2A targets from previous Omics data from several laboratories (Köhn, Kettenbach, Grana, Westermarck, Nilsson, and Saurin labs). In these comparisons the directionality of regulation i.e., whether PP2A inhibition causes hyper or hypophosphorylation of the target would be critical.

Specific comments:

3. Using log₂fold 0.58 (i.e., 1,5-fold) as a cut-off for differentially regulated proteins, about 40 % of phosphorylation sites were dephosphorylated, which is against the expected mode of action for a phosphatase inhibitor. What was the ratio between up and downregulated phosphosites when more commonly used log₂fold 1 was used as a cut-off, and how the authors explain that the small difference (60 vs. 40%) between up and downregulated phosphoproteins with the current cut-off? This is especially important as splicing factor phosphorylation regulation was seen much more strongly enriched among the hypophosphorylated targets i.e., in the unexpected group affected by phosphatase activity inhibition. In Fig. 1G, only very few splicing factors show hyperphosphorylation which further indicates that the impact of LB100 on splicing factor phosphorylation is not due to proposed mechanism of action i.e., PP2A inhibition.

4. Fig 3A and 3B: The proposed effect of LB100-elicited phosphorylation regulation on splicing factor targeting therapies is interesting. However, like for entire Fig.4, there is no data to back-up the hypothetical predictions. This is an important weakness of the paper. Testing the impact of LB100 on the response to SF3B1 and RBM39 targeting therapies would be the minimum level of validation required to justify these to hypothetical figures to be included in the manuscript.

Minor Comments:

5. Fig. 2F: It is not mentioned how the protein levels was measured

Referee #2:

The manuscript by Dias et al, describes a potentially interesting observation that the inhibition of PP2A through small molecule LB-100 results changing in the phosphorylation status of splicing factors, resulting in alternative spliced events, which may result in immunogenic neo-epitopes. These observations have been made following an RNAseq and proteomics experiment. While the findings presented in this manuscript have the possibility to have an important impact, I believe the data presented does not fully support the link between PP2A and splicing-derived neo-epitope production and immune sensitisation. Overall the data seems to be more correlative, than experimentally verified conclusions.

Major concerns/suggested experiments:

- It is unclear if the authors are describing AS mRNAs, or the actual neo-peptides from the ASEs themselves which overlap with the Lu et al datasets. There does not seem to be any analysis performed which identifies potential neo-peptides generated from ASEs in the manuscript. For example, computational pipelines exist, which can identify and predict potential binding affinity to MHC-I molecules of neo-peptides from RNAseq experiments. (PMID: 30114007). The neo-peptides identified from the RNAseq should be compared with IP/MS results from Lu et al.
- IP-MHC-I/MS should be performed to validate and identify LB-100 derived presented neo-peptides.
- The link to splicing factor phosphorylation changes and the observed ASEs identified by RNAseq is not particularly strong. Given that the phosphorylation changes are both hypo- and hyper-, is there conclusive evidence which splicing factor or factors have lost activity? The comparison with SF3B1 KD cells from a previous study is not sufficient. Does the deletion of the domains in which the clustered changes in phosphorylation sites of SF3B1/RBM39/SRSF1 result in immune activation/neo-peptide production?
- The neo-peptides from ATM, PARPBP and HNRNPM should be tested for immunogenicity.
- Given the authors have identified SF3B1 and RBM39 as primary candidates, a dual PladB/Indisalum treatment should result in an even more significant overlap of ASEs with LB-100?

Referee #3:

The paper by Dias et al explores phosphoproteome changes in colorectal adenocarcinoma cell lines upon treatments with PP2A phosphatase inhibitor LB-100. The authors identify differential phosphorylation (hyper- and hypo-) in various groups of proteins, particularly components of splicing machinery. In agreement, widespread changes in alternative splicing were detected by RNA-seq. These changes affected DNA repair regulators and were predicted to be source of neoantigens, potentially rendering cells sensitive to immune modulators and genotoxic agents. The study provides rationale for clinical potential of LB-100.

The paper is informative and will be of interest. I have the following comments and questions particularly regarding analyses and interpretation of the data:

Major comments:

1) Abstract states: "We report an unanticipated sensitivity of the splicing machinery to phosphorylation changes in response to PP2A inhibition". PP2A was reported to regulate splicing via dephosphorylation of the core spliceosome components SF3B1 and CDC5L (Shi et al, Mol Cell, 2006 doi: 10.1016/j.molcel.2006.07.022). This paper should be cited and discussed in the main text and expression "unanticipated" deleted in the abstract.

2) Text in the result section indicates that cells were treated for mass spec analyses with 4µl of LB-100 for 12h, but Figures and Materials and Methods section show 8h. Please correct. I am also missing information how/why these time points and concentration of the drug were selected. This information is important to reduce possibility of secondary and off-target affects.

3) In Fig.1g SRSF1 is shown to be hypophosphorylated (blue color), but in Supplemental Fig.3a three residues are hypo- and three hyperphosphorylated. Similarly, SF3B1 is shown to be about equally mixed in hyperphosphorylated and hypophosphorylated residues (Fig. 3a), but is shown mostly hypophosphorylated in Fig. 1g (mostly in blue color). These discrepancies should be clarified and data analyses and data presentation in interaction maps (Fig. 1g) and individual proteins (Fig. 3a and Sup Fig. 3a) explained. At minimum, it should be discussed why SF3B1 is dephosphorylated on 11 residues and hyperphosphorylated at 7 after PP2A inhibition when PP2A was shown to be a SF3B1 phosphatase (Shi et al Mol Cell, 2006 doi: 10.1016/j.molcel.2006.07.022). Phosphorylation status of SF3B1 (Girard et al, Nat Comm 2012, doi: 10.1038/ncomms1998. Shi

et al, 2006, Mol Cell doi: 10.1016/j.molcel.2006.07.022) should be checked by western blotting after treatment with different concentrations of the drug at various time points. This experiment may help clarify discrepancy with conclusions of Shi et al, Mol Cell, 2006.

4) Fig. 1g. and corresponding text in the result section: hnRNPs are hyperphosphorylated (thus potentially a direct target of the drug?) and most of the SR proteins are hypophosphorylated. The phosphorylation changes in these groups of proteins could potentially explain observed changes in the alternative splicing. Therefore, phosphorylation changes in these groups of proteins should be properly discussed in the text.

5) GO analyses of the hyperphosphorylated proteins (Fig. 1d) shows effect on mitosis-associated functions as a top hit. As these proteins could be potentially primary target of LB-100, and mitotic/cell cycle changes are of potential clinical targeting, this should be discussed and relevant mitosis-associated proteins specified in the text.

6) Fig. 4a, d; supplemental Fig. 4a: Protein levels of some of the candidate transcripts should be checked by Western blotting (ATM, PARPBP etc). Such a confirmation would significantly substantiate conclusions of the study.

Minor Comments:

1) Fig. 2c and 2g-mislabelled panel g for c

2) Fig. 2f, supplemental Fig. 2a, b, c, d: differences in violin plots (Fig. 2f) are not visible, specifically in comparison to violin plots in supplemental Fig. 2a,b,c,d.

3)Fig. 3b: Ser217 in RBM39 mentioned in the text is not shown in the figure.

We thank the reviewers for taking the time to review our manuscript and their insightful comments and suggestions. We made our best effort to implement whatever was feasible in the timeframe for revision. We believe it improved the work significantly.

Answer to the comments of the referees

Referee #1 (comments to authors):

The manuscript by Dias and co-workers analyzed differentially expressed proteomes and phosphoproteomes in response to 12-hour treatment with broad-specificity serine threonine phosphatase inhibitor LB100. LB100 and its derivative LB102 are in clinical development as cancer therapies, and they synergize with genotoxic therapies and induce anticancer immune responses. The work by Dias et al., is technically sound and the results interesting, but the authors totally neglect existing literature that none of the cantharidin derivatives, including LB100, cannot be considered as selective PP2A inhibitors (PMID: 27002182, 17200551 etc.). In fact, LB100 was recently shown to be equally efficient inhibitor of another serine threonine phosphatase PP5 than PP2A (PMID: 30679389). Further, the anti-tumor effects of LB100 that the authors call surprising, are not surprising as also other broad-specificity serine threonine phosphatase inhibitors such as okadaic acid are cytotoxic to cancer cells, and because the target of LB100, PP5, is oncogenic (PMID: 37527661, 29805615). Therefore, the current manuscript is severely misleading, and there is no basis for the main conclusions that the observed effects by LB100 would be mediated by selective inhibition of PP2A. In fact, none of the LB100 papers published so far have been able to convincingly demonstrate PP2A selectiveness of the reported effects.

There are two alternative strategies to solve this major issue:

1. Rewriting the entire manuscript so that all mentions related to PP2A are removed, and the effects are described solely as pharmacological responses to LB100. This must be supplemented by careful discussion of the existing literature about lack of the non-selectivity of LB100 against serine threonine phosphatases. This strategy would include NOT starting the introduction with PP2A, but carefully introducing the evidence that LB equally well inhibits PP5 and PP2A and the expected cancer selective effects of inhibitions of these phosphatases.
2. Providing solid validation data demonstrating that the major conclusions can be recapitulated by direct inhibition of PP2A by for example siRNA or CRISPR/Cas9. Parallel to that, authors should compare the identified splicing targets to known PP2A targets from previous Omics data from several laboratories (Köhn, Kettenbach, Grana, Westermarck, Nilsson, and Saurin labs). In these comparisons the directionality of regulation i.e., whether PP2A inhibition causes hyper or hypophosphorylation of the target would be critical.

Answer:

We are glad the reviewer finds our manuscript "*technically sound*." While we agree that LB-100, despite its widely described effect on PP2A (e. g. PMIDs: 25897893, 29199006, 28588271, and others) can also inhibit PP5, we believe that the critical aspect of our findings is the previously unrecognized effect of LB-100 on modulating splicing and creating

neopeptides. In light of this, , we have followed the suggestion 1 from the reviewer and re-wrote the manuscript, clarifying that we focus on effect of LB-100 without strong argumentation whether its mechanism of action is solely dependent on PP2A inhibition. As also suggested by the Reviewer, we have reanalyzed the previously published omics datasets of PP2A inhibited cells, uncovering a concordance in LB-100 mis-phosphorylated targets and PP2A-KD rewired phosphoproteome (Expanded View Figure 1A-C). Critically, these data on PP2A inhibited cells exhibit many of the differentially phosphorylated splicing factors we have identified (Expanded View Figure 1B, C), arguing that observed effects of LB-100 are, at least in part, mediated through PP2A modulation. This degree of specificity may be also supported by the observation that other target of LB-100, PPP5C, is expressed at approximately ~3 times lower levels than PP2AC in cancer tissues, including colorectal adenocarcinoma (Protein Atlas). Nevertheless, the manuscript now refers primarily to LB-100 as a phosphatase inhibitor and only refers to similarities of LB-100 and PP2A inhibition based on data.

Answer to specific comments:

3. Using log2fold 0.58 (i.e., 1,5-fold) as a cut-off for differentially regulated proteins, about 40 % of phosphorylation sites were dephosphorylated, which is against the expected mode of action for a phosphatase inhibitor. What was the ratio between up and downregulated phosphosites when more commonly used log2fold 1 was used as a cut-off, and how the authors explain that the small difference (60 vs. 40%) between up and downregulated phosphoproteins with the current cut-off? This is especially important as splicing factor phosphorylation regulation was seen much more strongly enriched among the hypophosphorylated targets i.e., in the unexpected group affected by phosphatase activity inhibition. In Fig. 1G, only very few splicing factors show hyperphosphorylation which further indicates that the impact of LB100 on splicing factor phosphorylation is not due to proposed mechanism of action i.e., PP2A inhibition.

Answer:

Our proposed model for the presence of the bi-directional changes in phosphorylation patterns in response to LB-100 is that the treatment affects kinases and/or phosphatases downstream of LB-100. However, rather than understanding in depth molecular effectors of LB-100, we focus on therapeutic implications of the findings, exploring the possibility of novel splicing vulnerabilities in LB-100 treated cancers.

Following Reviewer's suggestion, we have also performed an analysis of our phosphoproteomics dataset adjusting the cut-off to log2FC > 1. It indeed slightly shifted the number of affected phosphorylation sites in favor of hyperphosphorylation events (1598 hyperphosphorylated sites in 863 proteins versus 498 hypophosphorylated sites in 326 proteins enriched for splicing factors). However, it had no impact on the overall conclusions of the manuscript. Therefore, we decided to keep previous cut-offs to ensure high sensitivity and robustness of the analysis.

4. Fig 3A and 3B: The proposed effect of LB100-elicited phosphorylation regulation on splicing factor targeting therapies is interesting. However, like for entire Fig.4, there is no data to back-up the hypothetical predictions. This is an important weakness of the paper. Testing the impact of LB100 on the response to SF3B1 and RBM39 targeting therapies would be the minimum

level of validation required to justify these to hypothetical figures to be included in the manuscript.

Answer:

We fully agree that addressing the effects of combinatorial use of LB-100 with splicing targeting drugs could strengthen the argument of the paper. We therefore followed the reviewer's suggestion and performed viability assays in cells treated with LB-100 together with selected inhibitors of RBM39 (indisulam), SF3B1 (pladienolide B) and the spliceosome nodes unrelated to differentially phosphorylated proteins identified in this study (isoginkgetin, small compound that interferes with tri-snRNP). Curiously, we observe cell line-specific sensitization of human colorectal carcinomas to combination of the LB-100 with RBM39 and SF3B1 modulators, but not with an inhibitor of unrelated to LB-100 treatment tri-snRNP (Figure 4D and Expanded View Figure 4).

Referee #2 (comments to authors):

The manuscript by Dias et al, describes a potentially interesting observation that the inhibition of PP2A through small molecule LB-100 results changing in the phosphorylation status of splicing factors, resulting in alternative spliced events, which may result in immunogenic neo-epitopes. These observations have been made following an RNAseq and proteomics experiment. While the findings presented in this manuscript have the possibility to have an important impact, I believe the data presented does not fully support the link between PP2A and splicing-derived neo-epitope production and immune sensitisation. Overall the data seems to be more correlative, than experimentally verified conclusions.

Comment:

We are grateful to the Reviewer for acknowledging that our „*findings presented in the manuscript have the possibility to have an important impact*“. We also fully agree that providing evidence for production of LB-100 treatment-triggered neoepitopes would substantially strengthen the implications of our manuscript.

Major concerns/suggested experiments:

- It is unclear if the authors are describing AS mRNAs, or the actual neo-peptides from the ASEs themselves which overlap with the Lu et al datasets. There does not seem to be any analysis performed which identifies potential neo-peptides generated from ASEs in the manuscript. For example, computational pipelines exist, which can identify and predict potential binding affinity to MHC-I molecules of neo-peptides from RNAseq experiments. (PMID: 30114007). The neo-peptides identified from the RNAseq should be compared with IP/MS results from Lu et al.

- IP-MHC-I/MS should be performed to validate and identify LB-100 derived presented neo-peptides.

Answer:

As stated above, we agree with this important comment. We have therefore performed a quantitative immunopeptidomics experiment with LB-100 treated HT-29 colon adenocarcinoma cells. This have unearthed variety of peptides with neoantigen potential formed downstream to LB-100 (Figures 5 D-F and S5B, C). Notably, it seemed that LB-100-treated cells had a tendency toward increased presentation of unique immunopeptides compared to control cells (141 vs. 31). At the moment we cannot define how many of those were triggered by splicing alterations, likely because of the previously reported difficulty in mapping MHC-I bound peptides to the proteome that depends on the employed reference library (PMID: 28244318). However, we observe that 7 of alternatively spliced transcripts upon LB-100 could lead to neoantigen formation (Figure 5G). This percentage (5.1% of experimentally validated immunopeptides) is in agreement with previous report from Bradley lab (PMID: 34171309 – 7.6 to 8.1%, depending on the MHC haplotype). Furthermore, as suggested by the Reviewer, we have included a computational analysis of MHC I binding affinities of predicted, alternative splicing-derived neopeptides by employing NetMHCpan 4.1 pipeline.

- The link to splicing factor phosphorylation changes and the observed ASEs identified by RNAseq is not particularly strong. Given that the phosphorylation changes are both hypo- and hyper-, is there conclusive evidence which splicing factor or factors have lost activity? The comparison with SF3B1 KD cells from a previous study is not sufficient. Does the deletion of the domains in which the clustered changes in phosphorylation sites of SF3B1/RBM39/SRSF1 result in immune activation/neo-peptide production?

Answer:

An in depth understanding of the importance of specific phosphorylation events in splicing factors would be certainly valuable. However, given the breadth of this research and likely complexity of these interactions, we feel that delineating all of them would be beyond the scope of the presented manuscript. We would like to keep the main focus on determining reasons for the existence of previously poorly understood cellular outcomes in LB-100 treated cancers. Nonetheless in the response to the specific comment 4 of Reviewer #1, we have now included in revised Figure 4D an analysis of alternative splicing events shared between LB-100-stimulated cells and cells treated with splicing targeting drugs and/or knock-downs from previously published datasets.

- The neo-peptides from ATM, PARBP and HNRNPM should be tested for immunogenicity.

Answer:

ATM and PARBP are examples of alternatively spliced transcripts encoding critical regulators of DNA damage response, likely to be one of the reasons for increased genotoxic stress elicited by LB-100 that was reported previously (PMID: 25245035). We have clarified this point in the text, validating also the protein levels of ATM and Chk2 (no reliable PARBP antibody

was available) these regulators as requested by Reviewer #3 (please see below and Figure 3C). For HNRNPM and other potential neopeptide-generating transcripts that are alternatively spliced in LB-100 treated cells, we have performed *in silico* prediction of the potentially generated neoantigens using established pipeline of NetMHCpan4.1 (Figure 5B). Together with the quantitative immunopeptidomics performed in LB-100-treated human colorectal adenocarcinoma cell line HT-29, these results are presented in revised Figure 5.

- Given the authors have identified SF3B1 and RBM39 as primary candidates, a dual PladB/Indisalum treatment should result in an even more significant overlap of ASEs with LB-100?

Answer:

This is a valid suggestion that could help to determine the potential applicability of combination therapies employing LB-100 in conjunction with splicing inhibitors. We have performed treatment with combinations of splicing inhibitors (pladienolide B to inhibit SF3B1, indisulam to target RBM39 and isoginkgetin as a drug not targeting LB-100-affected spliceosome components) in conjunction with LB-100. This analysis revealed synergy between SF3B1/RBM39 inhibitors and LB-100 for the SW-480 cell line but not for HT-29 (Figures 4E and EV4H-K).

Referee #3 (comments to authors):

The paper by Dias et al explores phosphoproteome changes in colorectal adenocarcinoma cell lines upon treatments with PP2A phosphatase inhibitor LB-100. The authors identify differential phosphorylation (hyper- and hypo-) in various groups of proteins, particularly components of splicing machinery. In agreement, widespread changes in alternative splicing were detected by RNA-seq. These changes affected DNA repair regulators and were predicted to be source of neoantigens, potentially rendering cells sensitive to immune modulators and genotoxic agents. The study provides rationale for clinical potential of LB-100. The paper is informative and will be of interest. I have the following comments and questions particularly regarding analyses and interpretation of the data:

Comment:

We would like to thank Reviewer for finding our paper *informative* and of *interest*. We are particularly glad that he/she agrees with us on what we believe is a main strength of our findings, that “*the study provides rationale for clinical potential of LB-100*”. We are grateful for all of specific comments, pointing out parts of the manuscript that require clarification or correction. We have included in our discussion a suggested important previous report showing dependency of SF3B1 phosphorylation on PP2A. We have further delineated in the text that our observations suggest that bi-directional changes in phosphorylation status of candidate splicing factors are likely secondary or indirect effects of PP2A inhibition. With that, we believe that it is not a major criticism for main findings of the paper, that reside mostly in explanation of previously reported but incompletely understood effects of LB-100 on function of cancer cells.

Major comments:

1) Abstract states: "We report an unanticipated sensitivity of the splicing machinery to phosphorylation changes in response to PP2A inhibition". PP2A was reported to regulate splicing via dephosphorylation of the core spliceosome components SF3B1 and CDC5L (Shi et al, Mol Cell, 2006 doi: 10.1016/j.molcel.2006.07.022). This paper should be cited and discussed in the main text and expression "unanticipated" deleted in the abstract.

Answer:

We have included discussion of the suggested paper in the text and have rewritten the abstract accordingly.

2) Text in the result section indicates that cells were treated for mass spec analyses with 4 μ l of LB-100 for 12h, but Figures and Materials and Methods section show 8h. Please correct. I am also missing information how/why these time points and concentration of the drug were selected. This information is important to reduce possibility of secondary and off-target effects.

Answer:

We have clarified and corrected this part in the text.

3) In Fig.1g SRSF1 is shown to be hypophosphorylated (blue color), but in Supplemental Fig.3a three residues are hypo- and three hyperphosphorylated. Similarly, SF3B1 is shown to be about equally mixed in hyperphosphorylated and hypophosphorylated residues (Fig. 3a), but is shown mostly hypophosphorylated in Fig. 1g (mostly in blue color). These discrepancies should be clarified and data analyses and data presentation in interaction maps (Fig. 1g) and individual proteins (Fig. 3a and Sup Fig. 3a) explained. At minimum, it should be discussed why SF3B1 is dephosphorylated on 11 residues and hyperphosphorylated at 7 after PP2A inhibition when PP2A was shown to be a SF3B1 phosphatase (Shi et al Mol Cell, 2006 doi: 10.1016/j.molcel.2006.07.022). Phosphorylation status of SF3B1 (Girard et al, Nat Comm 2012, doi: 10.1038/ncomms1998. Shi et al, 2006, Mol Cell doi: 10.1016/j.molcel.2006.07.022) should be checked by western blotting after treatment with different concentrations of the drug at various time points. This experiment may help clarify discrepancy with conclusions of Shi et al, Mol Cell, 2006.

Answer:

We would like to clarify that while Figure 1 shows only the results for significantly affected phosphosites, in Figure 4A, B and Expanded View Figure 4G, all detected sites are highlighted, with significant ones depicted in red. We have now improved data representation and descriptions to avoid confusion for future readers.

As suggested, we have tried to cross-validate our phosphoproteomic analysis with immunoblotting. However, none of the commercially available antibodies (p-Thr211; p-Thr-313) is specific for significantly regulated phosphosites affected by LB-100. Given time

constraints, we could not raise our own set of antibodies, and could not obtain meaningful results with antibody against total SF3B1, which disallowed addressing this comment.

4) 5) Fig. 1g. and corresponding text in the result section: hnRNPs are hyperphosphorylated (thus potentially a direct target of the drug?) and most of the SR proteins are hypophosphorylated. The phosphorylation changes in these groups of proteins could potentially explain observed changes in the alternative splicing. Therefore, phosphorylation changes in these groups of proteins should be properly discussed in the text.

GO analyses of the hyperphosphorylated proteins (Fig. 1d) shows effect on mitosis-associated functions as a top hit. As these proteins could be potentially primary target of LB-100, and mitotic/cell cycle changes are of potential clinical targeting, this should be discussed and relevant mitosis-associated proteins specified in the text.

Answer:

We have discussed the mitosis and cell cycle relevant genes in the text. Additionally, we have included in the manuscript the analysis of binding of differentially phosphorylated hnRNPs within alternatively spliced transcripts in response to LB-100, which is included in Expanded View Figure 4B-F.

6) Fig. 4a, d; supplemental Fig. 4a: Protein levels of some of the candidate transcripts should be checked by Western blotting (ATM, PARPBP etc). Such a confirmation would significantly substantiate conclusions of the study.

Answer:

We have performed immunoblotting analysis of selected, alternatively spliced regulators of DNA damage in response to LB-100 (ATM and Chk2), which is also a part of response to suggestions of Reviewer #2 described in Figure 3C.

Dear Dr. Ciesla,

Thank you for the submission of your revised manuscript to our editorial offices. I have now received the reports from the three referees that I asked to re-evaluate your study, you will find below. As you will see, the referees now supports the publication of the study in EMBO reports. Referees #1 and #3 has some remaining concerns and suggestions to improve the manuscript, I ask you to address in a final revised manuscript.

- We plan to publish your manuscript in the Report format (as it has less than 25000 characters and 5 main and EV figures). For a Scientific Report we require that results and discussion sections are combined in a single chapter called "Results & Discussion". Please do this for your manuscript. For more details, please refer to our guide to authors: <http://www.embopress.org/page/journal/14693178/authorguide#researcharticleguide>

- Please add up to 5 keywords to the manuscript and order the manuscript sections like this, only using these names: Title page - Abstract - Keywords - Introduction - Results & Discussion - Materials and Methods - Data availability section - Acknowledgements - Disclosure and Competing Interests Statement - References - Figure legends - Expanded View Figure legends

- Please remove the sections 'Lead Contact' and 'Materials availability' from the manuscript text file.

- Please remove the referee tokens from the 'Data availability section' (presently named 'Data and code availability') and make sure that the datasets are public latest on the publication date of the manuscript.

- Please change the names of the EV figures to "Figure EVx" instead of "Figure Sx".

- Fig. EV3 (presently S3) has a lot of empty space and if there is only one panel, it is not necessary to name this wit A. Please check.

- We updated our journal's competing interests policy in January 2022 and request authors to consider both actual and perceived competing interests. Please review the policy <https://www.embopress.org/competing-interests> and update your competing interests if necessary. Please name this section 'Disclosure and Competing Interests Statement' and put it after the Acknowledgements section.

Please also include here the statement that Reuven Agami is a member of the Advisory Editorial Board of EMBO reports and that this has no bearing on the editorial consideration of this article for publication.

- We now use CRediT to specify the contributions of each author in the journal submission system. CRediT replaces the author contribution section. Please use the free text box to provide more detailed descriptions and do NOT provide your final manuscript text file with an author contributions section. See also our guide to authors: <https://www.embopress.org/page/journal/14693178/authorguide#authorshipguidelines>

- There is an author name discrepancy: It is 'Matheus H. Dias' in the manuscript text file, but 'Matheus Dos Santos Dias' in the submission system. Please check.

- Please provide the complete affiliations for all authors. No affiliation for '2' is listed.

- Please make sure that the number "n" for how many independent experiments were performed, their nature (biological versus technical replicates), the bars and error bars (e.g. SEM, SD) and the test used to calculate p-values is indicated in the respective figure legends (for main, EV and Appendix figures) of the final revised manuscript. Please also check that all the p-values are explained in the legend, and that these fit to those shown in the figure. Please provide statistical testing where applicable. Please avoid the phrase 'independent experiment', but clearly state if these were biological or technical replicates. Please also indicate (e.g. with n.s.) if testing was performed, but the differences are not significant. In case n=2, please show the data as separate datapoints without error bars and statistics. See also:

<http://www.embopress.org/page/journal/14693178/authorguide#statisticalanalysis>

If n<5, please show single datapoints for diagrams. It seems, that for some diagrams n.s. is missing or that they show no or only partial statistics (see e.g. 4B, 5B, EV4C, S2 and S5). Please check. Moreover:

- Please note that legend for figure 2g is incorrectly labelled as 2c in the legend. This needs to be rectified.

- Please indicate the statistical test used for data analysis in the legends of figures 1b-e; 3a; 5f; EV 1b.

- Please note that in figures 2d, f; EV 1e, g; there is a mismatch between the annotated p values in the figure legend and the

annotated p values in the figure file that should be corrected.

- Please note that for the figure 2g, EV 1a, EV 2a-d; p-values and statistical tests are indicated in the legends. However, comparison for the same, "****/**/*" has not been represented in the figures. Please rectify this in the figures or legends as applicable.
- Please note that the box plots need to be defined in terms of minima, maxima, centre, bounds of box and whiskers, and percentile in the legends of figures 2d-e.
- Please note that information related to n is missing in the legends of figure 2f; EV 2a-d.

- Please add to each legend a 'Data Information' section explaining the statistics used or providing information regarding replicates and scales.

- Please use our reference format:

- Per journal policy, we do not allow 'data not shown', which is stated twice in the manuscript (page 8 and 9). All data referred to in the paper should be displayed in the main or Expanded View figures, or an Appendix. Thus, please add these data (or change the text accordingly if these data are not central to the study). See:

<https://www.embopress.org/page/journal/14693178/authorguide#unpublisheddata>

- Please upload a complete author checklist with your final submission, which you can download from our author guidelines (<https://www.embopress.org/page/journal/14693178/authorguide>). Please insert page numbers in the checklist to indicate where the requested information can be found in the manuscript. The completed author checklist will also be part of the RPF.

Please also follow our guidelines for any use of living organisms, and the respective reporting guidelines:

- Please make sure that all the funding information is also entered into the online submission system and that it is complete and similar to the one in the acknowledgement section of the manuscript text file. Presently, a grant from the Dutch Cancer Society and of the Dutch Ministry of Health, Welfare and Sport and by the OncoCode Institute, and a research grant from Lixte Biotechnology are only mentioned in the acknowledgements section. Please check.

- Tables EV1-EV6 are Datasets. Please upload these as dataset files, named Dataset EV1-EV6, and update their callouts accordingly. Please put a title and the legend on the first TAB of the excel files and remove the legends from the manuscripts text file. If some of these datasets are actually source data, please include them into the source data (see below) and do not upload them as datasets. In that case, please remove their callouts from the manuscript text.

- Thank you for providing the source data (SD). Please upload the SD for the main figures as one ZIPed folder per figure and the SD for EV figures grouped together in one ZIPed folder. If some of the datasets are SD, please include them in the SD folders and do not upload these as datasets. In that case, please remove their callouts from the manuscript text.

In addition, I would need from you:

Best,

Referee #1:

The revisions provided by the authors appropriately address most of the comments, and I appreciate the inclusion of new

bioinformatics and experimental data. Regarding my primary concern about LB100's target selectivity, the authors have revised the text in most parts accordingly. However, there are still a few unresolved issues that need attention:

1. Please ensure that when introducing LB100's inhibition of both PP2A and PP5, original references demonstrating this are cited. Additionally, citing evidence supporting the oncogenic nature of PP5 would help clarify LB100's anti-oncogenic effects. Some sentences still suggest that LB100's effects are mediated through PP2A, despite no evidence provided for PP2A selectivity in this report. For instance, "Surprisingly, inhibition of PP2A with LB-100 has also exhibited anti-cancer effects, especially when used in combination with radiotherapy or specific chemotherapy drugs." Since there's no evidence for PP2A selectivity, please remove any mention of PP2A.
2. It is very important that authors now present data showing how LB100 regulates phosphosites that overlap with PP2A-regulated sites from previous studies. However, it's necessary to clarify how these two datasets were generated and which was used in each analysis in EV1. Although all figure panels indicate the use of both studies, the information in the text and figure legend is inconsistent. For example while only the Hoerman et al. study is cited in the results section, only the Kauko et al. study is mentioned in the discussion. Similar inconsistencies can be found from the figure legend.

Referee #2:

The authors have performed experiments that strengthen their conclusions. I believe the revised manuscript has been significantly improved.

Referee #3:

Authors addressed some of my questions, but they did not corrected/addressed several points I have asked for, even though they claimed to do so in the response:

- ad question 1) "unanticipated" is still in the abstract
- 2)no information why selected concentration of LB-100 was used was added to the MS
- 3) Four significantly changed phosphorylation sites in SF3B1 were marked in "red". However, all of them are decreased (based on the legend in the figure). What is then the point of highlighting SF3B1 as target of phosphatase inhibitor LB-100?
- 4) I have not found any cell cycle/mitotic gene discussed in the text.

We thank the reviewers for taking the time to review our manuscript and for indicating remaining issues in the manuscript. We provide specific answers below.

Referee#1:

The revisions provided by the authors appropriately address most of the comments, and I appreciate the inclusion of new bioinformatics and experimental data. Regarding my primary concern about LB100's target selectivity, the authors have revised the text in most parts accordingly. However, there are still a few unresolved issues that need attention:

1. Please ensure that when introducing LB100's inhibition of both PP2A and PP5, original references demonstrating this are cited. Additionally, citing evidence supporting the oncogenic nature of PP5 would help clarify LB100's anti-oncogenic effects. Some sentences still suggest that LB100's effects are mediated through PP2A, despite no evidence provided for PP2A selectivity in this report. For instance, "Surprisingly, inhibition of PP2A with LB-100 has also exhibited anti-cancer effects, especially when used in combination with radiotherapy or specific chemotherapy drugs." Since there's no evidence for PP2A selectivity, please remove any mention of PP2A.

We have now included mention of PP5 involvement in cancer (p. 2), cited d'Arcy and colleagues' paper, and went through the text to amend the inconsistencies in mentioning the effect of LB-100 on PP2A/PP5.

2. It is very important that authors now present data showing how LB100 regulates phosphosites that overlap with PP2A-regulated sites from previous studies. However, it's necessary to clarify how these two datasets were generated and which was used in each analysis in EV1. Although all figure panels indicate the use of both studies, the information in the text and figure legend is inconsistent. For example, while only the Hoerman et al. study is cited in the results section, only the Kauko et al. study is mentioned in the discussion. Similar inconsistencies can be found from the figure legend.

This comment was addressed on page 4 of the revised manuscript. We have also unified the citations as suggested by the Reviewer.

Referee#2:

The authors have performed experiments that strengthen their conclusions. I believe the revised manuscript has been significantly improved.

Referee#3:

Authors addressed some of my questions, but they did not correct/addressed several points I have asked for, even though they claimed to do so in the response:

ad question 1) "unanticipated" is still in the abstract

It is now removed from the Abstract (p. 2).

2)no information why selected concentration of LB-100 was used was added to the MS
Answer:

Based on our preliminary data and previous experience with employing these colon cancer models, 4 μM of LB-100 was a concentration inducing considerable biological effects (e. g. reduced cell viability, transcriptional changes, activation of specific signalling pathways, and rewiring of phosphosites) but not overly toxic, allowing for the eventual investigation of drug combinations and studying molecular mechanisms of action not related to the direct dysfunction of the cells.

3) Four significantly changed phosphorylation sites in SF3B1 were marked in "red". However, all of them are decreased (based on the legend in the figure). What is then the point of highlighting SF3B1 as target of phosphatase inhibitor LB-100?

Answer:

As we propose in the manuscript, we believe that hypo-phosphorylated sites are likely a secondary biological consequence of LB-100 rather than a direct result of the inhibition of phosphatases activity on these particular sites.

4) I have not found any cell cycle/mitotic gene discussed in the text.
It is now mentioned on p. 4.

Dr. Maciej Ciesla
IMol Polish Academy of Sciences
Flisa 6
Warszawa, mazowieckie 02-247
Poland

Dear Dr. Ciesla,

I am very pleased to accept your manuscript for publication in the next available issue of EMBO reports. Thank you for your contribution to our journal.

Yours sincerely,
